# Large Language Model as Attributed Training Data Generator: A Tale of Diversity and Bias

**Yue Yu**[1,*] **Yuchen Zhuang**[1,*] **Jieyu Zhang**[2,*] **Yu Meng**[3],
**Alexander Ratner**[2], **Ranjay Krishna**[2], **Jiaming Shen**[4], **Chao Zhang**[1]
[1] Georgia Tech    [2] University of Washington    [3] UIUC    [4] Google Research
{yueyu, yczhuang, chaozhang}@gatech.edu, yumeng5@illinois.edu
{jieyuz2, ajratner, ranjay}@cs.washington.edu, jmshen@google.com

## Abstract

Large language models (LLMs) have been recently leveraged as training data generators for various natural language processing (NLP) tasks. While previous research has explored different approaches to training models using generated data, they generally rely on simple class-conditional prompts, which may limit the diversity of the generated data and inherit systematic biases of LLM. Thus, we investigate training data generation with diversely attributed prompts (e.g., specifying attributes like length and style), which have the potential to yield diverse and attributed generated data. Our investigation focuses on datasets with high cardinality and diverse domains, wherein we demonstrate that attributed prompts outperform simple class-conditional prompts in terms of the resulting model's performance. Additionally, we present a comprehensive empirical study on data generation encompassing vital aspects like bias, diversity, and efficiency, and highlight three key observations: firstly, synthetic datasets generated by simple prompts exhibit significant biases, such as regional bias; secondly, attribute diversity plays a pivotal role in enhancing model performance; lastly, attributed prompts achieve the performance of simple class-conditional prompts while utilizing only 5% of the querying cost of ChatGPT associated with the latter[2].

## 1 Introduction

Large language models (LLMs) have demonstrated exceptional performance across a broad range of NLP tasks [3, 39, 28, 37, 38, 58]. In recent research, LLMs have been proposed as training data generators, particularly for text classification, aiming to alleviate the need for task-specific data and annotations [54, 15, 55, 31, 57, 6]. While these efforts have showcased the effectiveness of LLMs as data generators, the focus has primarily been on advancing the training stage, where the generated data are utilized to train task-specific models, leaving the upstream data generation process relatively unexplored. Notably, the prevailing approach employs a simple class-conditional prompt for querying LLMs during data generation, potentially limiting the diversity of the generated data [6, 51] and inheriting systematic biases inherent in LLMs [13, 60, 24]. We refer to this simple class-conditional prompt as SimPrompt, providing an example in Table 1.

In this work, we ground the LLM to ChatGPT [38] for its ability to generate high-quality, human-like text, and consider four challenging topic classification tasks with high cardinality from various domains. Our investigation primarily revolves around assessing the bias and diversity present within the generated training set through the lens of *data attributes*. In particular, data attributes encompass multiple attribute dimensions and their corresponding attribute values, where the latter represent

---

[*]These authors contributed equally to this work.

[2]The data and code are available on https://github.com/yueyu1030/AttrPrompt.

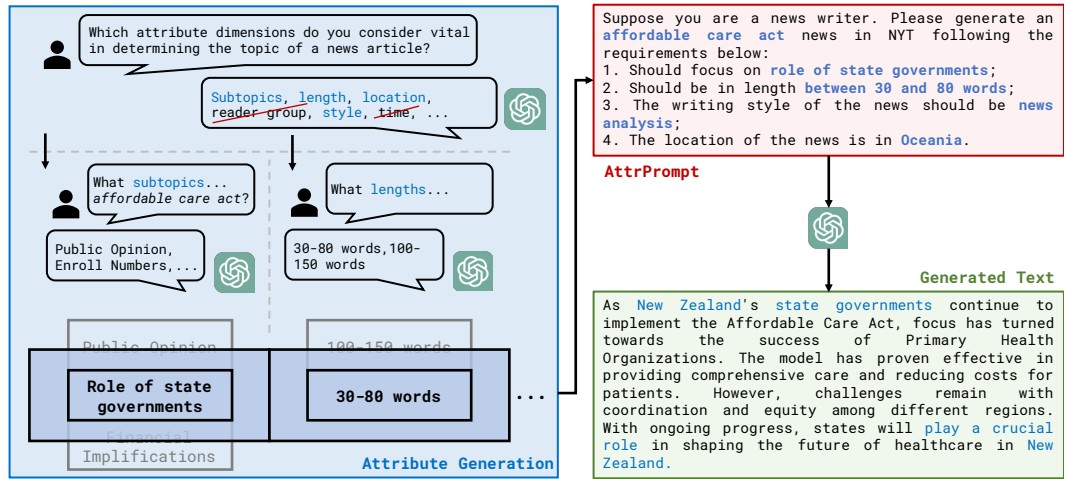

Figure 1: The overall workflow of AttrPrompt.

Table 1: Prompt template for the NYT news dataset.

| Method | Prompt |
|---|---|
| SimPrompt | Suppose you are a news writer. Please generate a {topic-class} news in NYT. |
| AttrPrompt | Suppose you are a news writer. Please generate a {topic-class} news in NYT following the requirements below:
1. Should focus on {subtopic};
2. Should be in length between {length:min-words} and {length:max-words} words;
3. The writing style of the news should be {style};
4. The location of the news should be in {location}. |

possible instantiations of the former. For example, an attribute value such as "*shorter than 200 words*" could serve as an instantiation of the attribute dimension "*length*".

On one hand, we employ a trained attribute classifier to examine the *attribute bias* present in the dataset generated using SimPrompt. When analyzing the "*location*" attribute in the NYT news dataset, we observe a striking bias towards "*North America*" in the predicted values of the generated data, accounting for a significant majority (68.01%). In contrast, instances associated with "*Africa*" are remarkably rare, comprising only 0.69% of the dataset (100 times less prevalent than "*North America*"). This regional bias exhibited in the generated dataset can pose substantial challenges when constructing reliable machine learning models [27, 4].

On the other hand, we explore the influence of *attribute diversity* on the downstream model performance. Specifically, we leverage ChatGPT to generate attributed data by incorporating desired attributes as constraints in the prompts. By comparing the performance of models trained on datasets generated using prompts with random attributes against those with fixed attributes, we observe a substantial underperformance of the latter, uncovering the importance of attribute diversity of the generated dataset. The workflow of AttrPrompt is shown in Figure 1.

To alleviate attribute biases and enhance the attribute diversity of the generated data, we propose to generate data with diversely attributed prompts. For a given classification task, we start by identifying attribute dimensions and their corresponding attribute values in an interactive, semi-automated process facilitated by the LLM. Subsequently, we generate diverse prompts by combining attributes randomly, replacing the simple class-conditional prompt typically used for querying data from the LLM. We refer to these diversely attributed prompts as AttrPrompt. An example of such prompts can be found in Table 1, where the LLM is instructed to generate training data based on attributes such as location and style.

On the four classification tasks, we empirically evaluate the generated datasets by measuring the performance of models trained using two scenarios: 1) solely on the generated dataset, and 2) on a merged dataset comprising the real training set and the generated set. In both scenarios, the dataset generated with AttrPrompt significantly outperforms its counterpart generated with SimPrompt. Furthermore, we demonstrate the superiority of AttrPrompt over SimPrompt in terms of data/budget

efficiency and compatibility with different model sizes/various LLM-as-training-data-generator approaches. Notably, AttrPrompt achieves the performance of SimPrompt while utilizing only 5% of the querying cost of ChatGPT associated with SimPrompt. Lastly, we extend the LLM-as-training-data-generator paradigm to the more challenging multi-label classification tasks for the first time, and AttrPrompt outperforms SimPrompt across all evaluation metrics.

## 2 Related Work

**LLMs as Training Data Generators.** With the remarkable success of large language models (LLMs), researchers have recently attempted to leverage them as training data generators. Such applications include generating tabular data [2], medical dialogue [7], sentence pairs [47], instruction data [40, 50, 53, 48], *etc.*. Among these applications, we anchor on training data generation for topic classification in a zero-shot setting where no labeled data is available. In this direction, existing approaches typically use simple class-conditional prompts while focusing on mitigating low-quality issues after generation. In particular, SuperGen [31] and ZeroGen [54] took initial steps to explore using LLM as a training data generator for text classification with simple class-conditional prompts, and leveraged additional noise robust learning techniques [26, 36, 52] to deal with the low-quality issue of the generated data. SunGen [15] reweights the generated data during training with learned data quality weight, and ProGen [55] leverages the model feedback to select highly influential generated data which then serve as labeled examples for generation. In this work, we instead explore attributed prompts to reduce the issue of low informativeness and redundancy, which can be readily incorporated into the existing systems mentioned above. Notably, Chen et al. [6] also explores prompts to advance the data generation process, yet it adopts soft prompts and requires a white-box LLM and seed examples to tune them. In contrast, our method is applicable to black-box LLMs and even LLM APIs (*e.g.*, ChatGPT) and does not rely on any labeled examples. Similarly, WANLI [29] also considers human-AI collaboration for creating more challenging training data, but requires an initial dataset and a strong task model. Instead, we aim to generate training data without any initial dataset or a pre-existing task model, which allows us to effectively handle resource-limited scenarios.

**Discrete Prompt Optimization.** Several works attempt to optimize discrete prompts for querying LLMs, such as prompt paraphrasing [22], prompt editing [42], or prompt searching [11]. Recently, [45, 59, 35] use large language models to optimize prompts. More related to us, [34] reframe prompts by decomposing a complex task instruction into multiple simple ones. However, these approaches mainly focus on the *inference* stage for directly predicting the answer and may rely on additional labeled examples for validation. Our focus is on an orthogonal setting, optimizing prompts for LLMs with attributes to diversify the generated training data. This approach improves the model's overall performance without the need for additional labeled examples.

## 3 Large Language Model as Attributed Training Data Generator

### 3.1 Datasets

While previous research has primarily focused on binary classification datasets [54, 31, 55] or datasets containing a maximum of 14 classes [15, 57], the performance of LLM as a data generator for topic classification with high cardinality (*i.e.*, many topic classes) remains unclear. Thus, we consider the following datasets from various domains with the number of topics ranging from 23 to 50:

- **NYT** [32]: The NYT dataset comprises news articles that were authored and published by *The New York Times*. These articles are categorized into 26 fine-grained categories.

- **Amazon** [1]: The Amazon dataset contains customer reviews on products from Amazon's online store. It covers products from 23 different categories.

- **Reddit** [17]: The Reddit dataset consists of a vast collection of user-generated content from the popular social media platform Reddit. It encompasses a wide range of topics, discussions, and interactions among users across numerous communities.

- **StackExchange** [17]: The StackExchange dataset is a rich collection of structured data encompassing various online communities and knowledge-sharing platforms. It contains a vast array of questions, answers, comments, tags, and user interactions about specific technical problems.

Table 2: Statistics of datasets.

| Dataset | Domain | Task | # Train | # Valid | # Test | # Class | Imbalance Ratio |
|---------|--------|------|---------|---------|--------|---------|-----------------|
| Amazon [1] | Reviews | Multi-class | 15.0K | 0.2K | 1.2K | 23 | 155.6 |
| NYT [32] | News | Multi-class | 9.0K | 0.2K | 1.2K | 26 | 357.0 |
| Reddit [17] | Web | Multi-class | 26.6K | 0.2K | 2.3K | 45 | 447.4 |
| StackExchange [17] | Web | Multi-class | 27.0K | 0.3K | 2.5K | 50 | 1283.7 |

Table 3: Attribute dimensions and values. Attributes with an asterisk* are class-dependent attributes.

| Dataset | # configurations / class | Attribute dimension | Attribute value |
|---------|--------------------------|---------------------|-----------------|
| NYT | 600 | Subtopic* | Appendix H.1.1 |
| | | Location | Asia, North America, South America, Africa, Oceania, Europe |
| | | Writing Style | Investigative journalism, Op-Eds, Feature writing, News analysis, Profiles and interviews |
| | | Length | short (30-80 words); long (100-150 words) |
| Amazon | 1000 | Product Brands* | Appendix H.2.1 |
| | | Product Names* | Appendix H.2.2 |
| | | Usage Experience | Worst, Bad, Average, Good, Excellent |
| | | Writing Style | Detailed Review; Comparative Review; Pros and Cons Review; Recommendation Review |
| | | Length | short (30-80 words); long (100-150 words) |
| Reddit | 500 | Resources* | Appendix H.3.1 |
| | | Experience* | Appendix H.3.2 |
| | | Writing Style | Informative/Educational; Entertaining/Funny; Discussion; Storytelling; Help/Advice |
| | | Length | short (30-80 words); long (100-150 words) |
| StackExchange | 400 | Scenario* | Appendix H.4.1 |
| | | Technical Depth | Beginner; Intermediate; Advanced; Expert |
| | | Writing Style | Specific; Comparative; Problem-Solution; Troubleshooting; Tutorial |
| | | Length | short (30-80 words); long (100-150 words) |

We summarize the statistics of used dataset in Table 2, from which we can see that the involved datasets not only have high cardinality but also come with high imbalance ratio, *i.e.*, the ratio of the sample size of the majority class to that of the minority class, which reflects the long-tail class issue in real applications.

## 3.2 Attributes

Our initial step involves identifying various types of data attributes (or metadata) that can be manipulated to generate attributed data samples. To facilitate this process, we employ ChatGPT to help establish both attribute dimensions and attribute values. Specifically, we begin by engaging ChatGPT in generating essential attribute dimensions. This is achieved by posing questions such as "*Which attribute dimensions do you consider vital in determining the topic of a news article?*" for the NYT dataset, resulting in responses like "*subtopics, length, location, reader group, style, time*". Then, we manually select the attribute dimensions of the highest quality that best suit the dataset. Similarly, we prompt ChatGPT (the prompt format is listed in Appendix G) to suggest potential attribute values within each attribute dimension and choose high-quality candidates.

**Attribute dimensions and values.** There are two types of attribute dimensions: *class-independent* attributes and *class-dependent* attributes. Class-independent attributes, such as "*length*", remain unchanged across different classes, while class-dependent attributes, like "*subtopic*", have varying attribute values for each class. We list attribute dimensions and values for all datasets in Table 3. These data attributes provide a human-manipulable interface for generating attributed data. In this study, we explore the potential of leveraging attributes to enhance the data generation process, while leaving the search for the optimal data attributes for a specific task to future work.

**Class-dependent attribute value filtering.** When dealing with class-dependent attributes, it is crucial to ensure that their attribute values are specifically associated with the corresponding class to avoid ambiguity and potential connections to multiple classes. For example, in the case of the "*economy*" class in the NYT dataset, a candidate attribute value generated by ChatGPT for the "*subtopic*" could be "*effect of trade tariffs on manufacturing companies*", which is also relevant to the "*international business*" class in the NYT. This overlap may introduce ambiguity in the generated data. To address this issue, we employ a filtering process called Class-Dependent Attribute Value Filtering (CAF). First, we query ChatGPT for the top-5 similar classes and then check with ChatGPT whether each class-dependent attribute value is related to these top-5 similar classes. Then, if the answer is positive which indicates a potential ambiguity, we remove that attribute value for the specific class.

### 3.3 Data generation and model training

Given the data attributes, one could prompt LLMs to generate data samples with diverse *attribute configurations*. For example, an attribute configuration for the "*federal budget*" class of the NYT dataset could be {"*subtopic*"="*defense spending*", "*length*"="*short:min-words=30, max-words=80*", "*style*"="*investigative journalism*", "*location*"="*North America*"}. In Table 3, we list the number of configurations per class, and one can further expand the number of configurations by adding more attribute dimensions and values. To generate attributed data samples, we prompt ChatGPT with random configurations. In particular, each time we generate a random configuration, complete a *prompt template* (see Table 1) with the generated configuration, and query ChatGPT with the completed prompt to collect generated data samples.

Throughout the experiments, we compare our method (AttrPrompt) against simple class-conditional prompt (SimPrompt, [54]) and the original training set of each dataset (Gold). For a fair comparison, we set the number of generated data the same as Gold for both AttrPrompt and SimPrompt. In principle, the generated dataset can be combined with any classifier (Sec. 5.4) and training techniques (Sec. 5.5); if not otherwise specified, we choose to fine-tune `BERT-base-uncased` [12] as the backbone and use the standard cross-entropy loss by default. For hyperparameter selection, we adhere to the recommendation in [41, 31] for SimPrompt and AttrPrompt, and *do not use the validation set* for model selection. Detailed hyperparameter configurations can be found in Appendix C.

## 4 Analysis of the Generated Data

### 4.1 A glimpse of the generated data

We begin by presenting examples of data generated by AttrPrompt and SimPrompt, and real data from the Gold set for the "*federal budget*" class in the NYT dataset (Table 4). It is evident that the data generated by ChatGPT exhibit high quality. Particularly, when comparing AttrPrompt to SimPrompt, we observe that AttrPrompt renders more diverse samples. This is because SimPrompt tends to generate news focused on the U.S., while AttrPrompt has the capability to generate news from various locations around the world.

Table 4: Data examples of different datasets: the "*federal budget*" class of the NYT dataset.

| Method | Example |
|---|---|
| Gold | 1. The emergency manager who was sent to reverse the fortunes of this financially troubled city asked some of its creditors on Friday to accept pennies on the dollar as he laid out his plan for tackling Detroit's staggering debt, kick starting negotiations that could determine whether the city is headed to bankruptcy court... 
 2. Saying that its debt could reach 45 billion by 2017 if Congress does not act, the Postal Service on Wednesday called on lawmakers to give it the flexibility to change its business model to keep itself solvent. 
 3. Governmental Affairs Committee, Patrick R. Donahoe, the postmaster general, asked Congress to give the Postal Service permission to run its own health plan for employees and retirees, modify a Congressional mandate that requires the agency to pay... |
| SimPrompt | 1. Washington D.C. The United States government has recently passed its federal budget for the next fiscal year, setting aside a whopping 4.8 trillion for spending. This is the largest budget in the country's history and reflects the government's commitment to improving the country's economy and infrastructural development. 
 2. WASHINGTON D.C. The Trump administration released its proposed budget for the federal government on Monday, calling for cuts to programs that provide assistance to low income Americans while increasing military spending. The proposed budget includes a 10 percent increase in defense spending, bringing the total military budget to 574 billion... 
 3. Washington D.C. The U.S. federal government has presented its budget for the fiscal year 2022, which includes a total of 6 trillion in spending and initiatives aimed at making the country more competitive and sustainable. The budget plan includes significant investments in infrastructure, healthcare, education, and renewable energy. |
| AttrPrompt | 1. In Germany, public opinion on federal budget issues has been influenced by The Federal Agency for Civic Education. Using in depth storytelling, this agency communicates budget policies to citizens, emphasizing their impact on society. Its efforts have led to greater engagement and awareness of budget issues among the public in recent years. 
 2. Op Eds The Government Spending Priorities in Mexico As the Mexican federal budget draws near, all eyes are on the government's spending priorities. One area of concern is the role of the Banco de Mexico in managing the country's finances. Many experts argue that the central bank should play a more active role in balancing the budget and stimulating economic growth... 
 3. TOKYO The recent federal budget proposal has triggered a wave of agency budget cuts and restructuring in Japan, with the Japan Foundation for Regional Development being one of the latest casualties. The foundation, which aims to promote regional development and revitalization, is set to have its budget slashed by 20 next year. |

Then, we show the vocabulary size of the generated dataset and the Gold dataset, which is a natural way to check the lexical diversity of datasets (Table 5). From the table, we can see that AttrPrompt has higher lexical diversity than SimPrompt in terms of both vocabulary size of the whole dataset (All in the table) and the averaged vocabulary size across classes (Class Avg. in the table). Yet, both have much smaller vocabulary sizes than the Gold, indicating there is still room for improvement of the ChatGPT's lexical diversity.

Lastly, we visualize the diversity of datasets via the distribution of cosine similarity of same-class text pairs (Figure 2), where the cosine similarity is calculated based on the embedding of Sentence-BERT [44], as well as including two additional metrics, namely *average pairwise sample similarity (APS)* and *inter-sample N-gram Frequency (INGF)* [33], as shown in table 6. We can see that the Gold dataset has the lowest cosine similarity, indicating that real data has the largest diversity. In contrast, the similarity between samples generated by SimPrompt is high Compared to SimPrompt,

Table 5: Comparison of the vocabulary size of different datasets.

| Method | NYT | | Amazon | | Reddit | | StackExchange | |
|---|---|---|---|---|---|---|---|---|
| | All | Class Avg. | All | Class Avg. | All | Class Avg. | All | Class Avg. |
| Gold | 70.8k | 11.3k | 44.7k | 6.64k | 50.8k | 4.62k | 52.3k | 3.60k |
| SimPrompt | 20.6k | 3.13k | 11.6k | 2.50k | 19.9k | 3.06k | 13.3k | 2.20k |
| AttrPrompt | 21.4k | 3.50k | 14.0k | 2.76k | 25.4k | 3.64k | 17.8k | 2.93k |

Table 6: Comparison of two quantitative metrics on diversity: the average pairwise sample similarity (APS) and inter-sample N-gram frequency (INGF) of different datasets. For APS, the *lower* stands for better diversity. For INGF, the *higher* stands for better diversity.

| Method | NYT | | | | Amazon | | | |
|---|---|---|---|---|---|---|---|---|
| | Inter-Class APS | Intra-Class APS | APS | INGF | Inter-Class APS | Intra-Class APS | APS | INGF |
| Gold | 0.098 | 0.358 | 0.122 | 7618.1 | 0.101 | 0.251 | 0.114 | 4992.1 |
| SimPrompt | 0.101 | 0.568 | 0.135 | 5277.2 | 0.207 | 0.620 | 0.241 | 2266.5 |
| AttrPrompt | 0.159 | 0.474 | 0.182 | 6688.6 | 0.225 | 0.483 | 0.246 | 2605.5 |
| | Reddit | | | | StackExchange | | | |
| | Inter-Class APS | Intra-Class APS | APS | INGF | Inter-Class APS | Intra-Class APS | APS | INGF |
| Gold | 0.044 | 0.261 | 0.054 | 9079.6 | 0.056 | 0.196 | 0.063 | 5492.4 |
| SimPrompt | 0.173 | 0.818 | 0.201 | 2697.8 | 0.282 | 0.804 | 0.302 | 2259.8 |
| AttrPrompt | 0.106 | 0.474 | 0.122 | 3994.5 | 0.105 | 0.375 | 0.114 | 2464.3 |

(a) NYT      (b) Amazon      (c) Reddit      (d) StackExchange

Figure 2: The distribution of cosine similarity of text pairs sampled from the same class.

dataset generated with AttrPrompt exhibits lower cosine similarity and the distribution is close to that of the Gold, which shows AttrPrompt could render more diverse data. Apart from the above automatic evaluation processes, we also conduct *human study* in Appendix E to manually evaluate the quality of the generated training data.

## 4.2 Is there attribute bias in real and generated data?

We study the attribute bias in both real dataset (Gold) and generated dataset of SimPrompt using dataset generated by AttrPrompt as a probe. In particular, we leverage the attributes associated with each data of AttrPrompt to train an *attribute classifier*, which is in turn used to make attribute predictions on Gold and SimPrompt dataset. Note that the attribute values associated with each data of AttrPrompt are not necessary the ground truth, yet since ChatGPT has shown remarkable performance in following instructions [39], the generated data could decently reflect the desired attributes and therefore the attribute classifier trained with them could partially reveal the underlying attribute distribution of tested dataset, *i.e.*, Gold and SimPrompt.

We pick the "*location*" attribute of the NYT data and visualize the distributions of the predicted "*location*" of Gold and SimPrompt in pie charts (Figure 3). One can see that both the Gold and SimPrompt dataset are largely biased towards "*North America*" in terms of the whole dataset (subfigures (a)&(e) in Figure 3). As to the "*location*" distribution for specific classes, we can see that Gold and SimPrompt are biased towards continents other than "*North America*". In contrast, with attributed prompts, the generated dataset of AttrPrompt comes with relatively balanced attribute distributions (Appendix F.2). While existing works using LLMs as data generators usually overlook the bias embedded in the generated data, we hope that this preliminary analysis could raise the attention of the community to the attribute bias behind the generated data of LLMs such as ChatGPT.

## 4.3 How important the attribute diversity is?

We investigate the impact of attribute diversity within AttrPrompt on model performance. Specifically, we conduct experiments by fixing one attribute dimension to a candidate value while keeping other

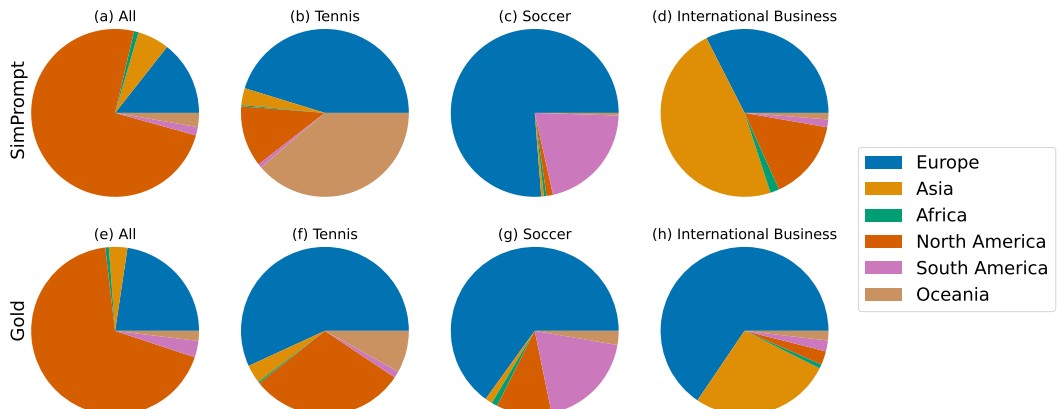

Figure 3: Pie charts of the distributions of "*location*" predicted by an attribute classifier for the NYT SimPrompt and Gold dataset. (a) and (e) are "*location*" distribution over the whole dataset, while others are for specific classes. Illustration on types of biases on other datasets can be found in Appendix F.1.

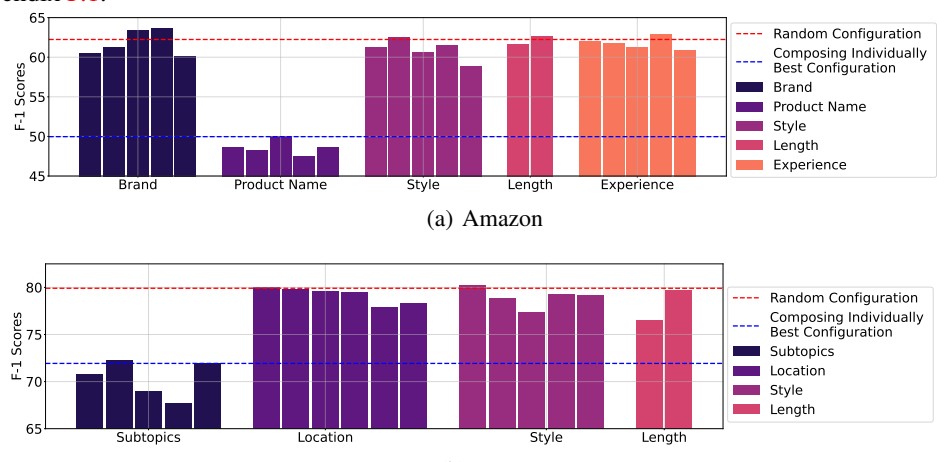

(a) Amazon

(b) NYT

Figure 4: Bar charts of model performance with different attribute configurations of AttrPrompt.

attribute values random. Then, we generate 50 data per class using such a one-fixed-others-random configuration to compose a dataset and evaluate the performance of the trained model. Note that for class-dependent attributes, we sample one value for each class and repeat it 5 times, since it is *computationally prohibitive* to enumerate all combinations of class-dependent attribute values. In Figure 4, each bar stands for a specific one-fixed-others-random configuration; compared to random configurations, most of one-fixed-others-random configurations result in a performance drop.

To further reduce the attribute diversity, we pick the attribute value with the best performance for each attribute dimension (the highest bar within each attribute dimension) and compose them to a single configuration (the dashed blue line). We can see that the dashed blue line is significantly worse than the random configuration, even though it is composed of individually best attribute values. This illustrates the importance and necessity of designing prompts with diverse attributes.

## 5 Experiments on the Trained Models

### 5.1 Training with generated data

We quantitatively evaluate the quality of generated datasets via the test performance of models trained with them. Apart from the AttrPrompt and the direct baseline SimPrompt, we include an additional baseline MetaPrompt [45] which leverage LLM to generate additional guidance information for improving upon SimPrompt. The details for MetaPrompt are shown in Appendix K. We also directly use ChatGPT as a zero-shot predictor for comparison. The results are in Table 7. Besides the test

performance, we include the cost of querying the ChatGPT per 1000 data in the table. From the results, we can draw the following conclusions. First, the AttrPrompt consistently renders better performance compared to the SimPrompt with a margin of 6–10 points[3]. Second, the class-dependent attribute value filter (CAF) is beneficial since the AttrPrompt outperforms its variant without CAF[4]. Third, although MetaPrompt [45] attempts to optimize the class-dependent prompt, they are less effective than AttrPrompt as they do not resolve the unique bottleneck (i.e. limited diversity) of the current prompting strategies for the training data generation task. Fourth, out of the four datasets, the AttrPrompt outperforms the LLM zero-shot method on three datasets in terms of accuracy, while for the F1 score, the AttrPrompt surpasses the LLM zero-shot on all the datasets; combined with the observation that the LLM zero-shot inference incurs much higher costs compared to data generation and the fact that the generated data is re-usable for training any model, we argue that for topic text classification generating training data could be a better practice of leveraging LLM than direct zero-shot inference. Lastly, in most cases, the generated data underperform the original training set, indicating that there is still room for future improvement. We conduct further studies in Appendix D.5 to illustrate the performance over different classes.

Table 7: Performance of the models trained with created datasets and the cost of constructing the datasets. The results are averaged over five runs. The gain of AttrPrompt has passed the statistical test with $p < 0.05$. We also include the performance and cost of using LLM as a zero-shot predictor.

| Method | NYT | | | Amazon | | | Reddit | | | StackExchange | | |
|---|---|---|---|---|---|---|---|---|---|---|---|---|
| | Acc. | F1 | Price/1k | Acc. | F1 | Price/1k | Acc. | F1 | Price/1k | Acc | F1 | Price/1k |
| LLM Zero-Shot | 74.16 | 69.84 | 5.44 | 59.55 | 54.56 | 2.11 | 67.00 | 56.66 | 2.89 | 44.70 | 43.80 | 3.12 |
| Gold | 83.80 | 81.02 | — | 82.23 | 81.12 | — | 84.22 | 83.38 | — | 67.56 | 63.28 | — |
| SimPrompt | 75.47 | 76.22 | 0.76 | 57.34 | 56.96 | 0.77 | 53.48 | 53.81 | 0.65 | 42.88 | 41.30 | 0.69 |
| MetaPrompt | 79.58 | 79.83 | 0.87 | 56.35 | 55.98 | 0.84 | 54.61 | 54.30 | 0.74 | 44.81 | 44.02 | 0.83 |
| AttrPrompt w/o CAF | 80.40 | 80.92 | 0.91 | 61.67 | 61.57 | 0.82 | 61.22 | 60.18 | 0.72 | 45.90 | 44.84 | 0.81 |
| AttrPrompt | 81.30 | 82.26 | 1.05 | 66.08 | 65.65 | 0.87 | 63.33 | 63.10 | 0.84 | 48.99 | 47.42 | 0.90 |

## 5.2 Augmenting existing dataset with generated data

Here, we merge the generated dataset and the original training set into a single training set, and then test the model performance when it is trained with the merged dataset to see whether the generated dataset can further improve model performance with the original training set available. We present the results in Table 8. From the table, we can see that the generated dataset is an effective complement to the original training set, since most of the generated datasets introduce performance gain when combined with the original training set, especially our AttrPrompt which leads to improvement for all the cases. This notable improvement with simple dataset merge may motivate future studies of more advanced ways of using the generated data as augmentations to boost existing dataset.

Table 8: Performance of the model trained with the original training set/augmented with the generated dataset. We present the performance gain/drop compared to using the original training set in green/red.

| Method | NYT | | Amazon | | Reddit | | StackExchange | |
|---|---|---|---|---|---|---|---|---|
| | Acc. | F1 | Acc. | F1 | Acc. | F1 | Acc | F1 |
| SimPrompt | 85.56 +1.76 | 86.34 +5.32 | 81.85 -0.38 | 80.23 -0.89 | 85.11 +0.89 | 84.88 +1.50 | 74.53 +6.97 | 74.23 +10.95 |
| Metaprompt | 87.14 +3.34 | 87.33 +6.31 | 82.12 -0.11 | 80.14 -0.98 | 84.71 +0.49 | 84.62 +1.24 | 76.02 +8.46 | 75.70 +12.42 |
| AttrPrompt w/o CAF | 85.71 +1.91 | 87.18 +6.16 | 82.24 +0.01 | 80.76 -0.36 | 85.86 +1.64 | 85.65 +2.27 | 75.16 +7.60 | 74.64 +11.36 |
| AttrPrompt | 87.47 +3.67 | 88.06 +7.04 | 83.95 +1.72 | 83.93 +2.81 | 86.08 +1.86 | 85.98 +2.60 | 76.86 +9.30 | 76.53 +13.25 |

## 5.3 The budget and sample efficiency of the generated data

Here, we aim to study two types of efficiency of the generated dataset, *i.e.*, budget efficiency and sample efficiency, on the model performance. We use Amazon and NYT as examples, and the results for other datasets are deferred to Appendix D.3. First, in Figure 5(a) and 5(b), we compare the budget efficiency of AttrPrompt against that of SimPrompt. Surprisingly, AttrPrompt only requires 5% of budget to be on par with or outperform SimPrompt with 100% of budget across all the datasets. This observation highlights the significance of diverse prompts in the training data generation process.

---

[3]In Appendix D.4 we show that simply increasing the temperature $t$ for SimPrompt does not significantly improve its performance.

[4]Some of the filtered attributes are exhibited in Appendix I.

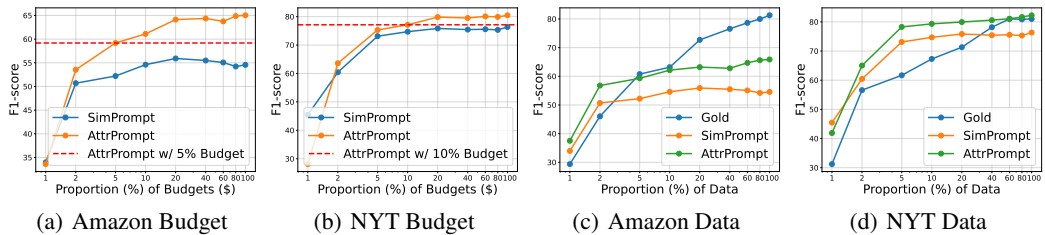

(a) Amazon Budget     (b) NYT Budget     (c) Amazon Data     (d) NYT Data

Figure 5: The comparisons on budget efficiency and data efficiency on Amazon and NYT datasets.

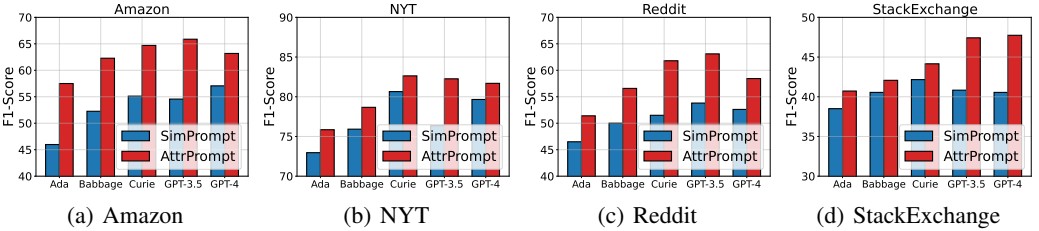

(a) Amazon     (b) NYT     (c) Reddit     (d) StackExchange

Figure 6: The barplot of performance with models of different parameter sizes. Note that due to budget limit, for GPT-4 model, the size of the generated dataset is only 10% of the full set thus the result is not directly comparable with other models.

Secondly, we examine the sample efficiency of Gold, SimPrompt, and AttrPrompt in Figure 5(c) and 5(d). While both SimPrompt and AttrPrompt exhibit better sample efficiency than Gold in the low-data regime, with superior performance when the dataset size is relatively small, Gold data shows better sample efficiency in the high-data regime. Overall, AttrPrompt renders better sample efficiency than SimPrompt, which suggests that increasing the diversity of the prompts could be an effective way to improve the unsatisfactory data scaling trend of using LLM as data generator [55].

### 5.4 The performance with respect to model parameter size

**Effect of the model size for LLM Generators.** To study the effect of different LLMs on Attr-Prompt, we use other instruction-finetuned GPT models as the generator, namely `text-ada-001` [39], `text-babbage-001` [39], `text-curie-001` [39], and GPT-4 [37] (due to budget constraints, we only generate a subset with 10% size of the original dataset). Under all settings, our model out-performs the direct baseline SimPrompt by a great margin. Besides, the performance is generally better with larger models, as they often have better instruction-following capabilities. In addition, an interesting finding is that for SimPrompt (but not for AttrPrompt), the average performance of using ChatGPT is worse than `text-curie-001`. This suggests that straightforward class-dependent prompts might not exploit the capabilities of LLMs as effectively as our proposed approaches.

**Effect of the model size for classifiers.** We experiment with other model choices in addition to the `BERT-base-uncased` [12] used throughout the paper. They are `TinyBERT` [23], `DistillBERT` [46], `DeBERTa-V3-base`, and `DeBERTa-V3-Large` [21] with parameter size from 14M to 435M[5]. We visualize the results in Figure 7. Overall, AttrPrompt outperforms SimPrompt by a large margin yet underperforms the Gold across different model choices. With a light-weighted backbone of 66M parameters, AttrPrompt can often outperform the SimPrompt trained with the model containing 435M parameters. This indicates that diversely attributed prompts could help close the performance gap between the Gold and simple class-conditional prompts, and such an improvement is robust to model parameter size.

### 5.5 Plugging AttrPrompt in existing approaches

In this section, we demonstrate that AttrPrompt can be painlessly integrated with prior zero-shot training data generation techniques. Table 9 shows the results for several recently proposed methods, which design additional techniques based on the noisy-robust loss to further reduce the effect of noisy

---

[5]Additionally, we also evaluate under the setting of linear probing [25] which trains a linear classifier over the frozen pretrained features. The results are deferred to Appendix D.1.

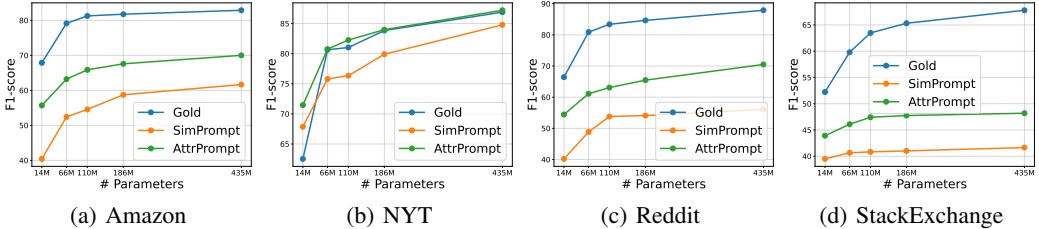

| (a) Amazon | (b) NYT | (c) Reddit | (d) StackExchange |

Figure 7: The performance curves with models of different parameter sizes.

labeled data [15, 31, 55], and leverage in-context examples for data generation [55]. Despite these approaches achieving notable performance gains on simple binary classification tasks, their gains become more marginal for fine-grained classification: the performance gain is less than 2% for all methods on two datasets. Instead, using AttrPrompt lead to consistent performance boosts (more than 5% in all cases) for those approaches, indicating that compared with label noise, *data diversity* is a more crucial bottleneck for existing dataset generation methods.

More interestingly, AttrPrompt even benefits dataset generation approaches that do not use LLMs. To demonstrate this, we use the LLM-generated contents (subtopics for NYT and product name for Amazon) to enrich the label names used in ReGen [57], a retrieval-based approach for training data generation. With the expanded label names, AttrPrompt largely improves (14%–26% absolute gain) the performance of ReGen on fine-grained classification tasks. These results justify the advantage of AttrPrompt for serving as a generic plug-in module for existing training data generation approaches.

Table 9: Performance comparison when AttrPrompt serves as a plug-in for existing approaches.

|  |  | SuperGen | SuperGen w/ AttrPrompt | SunGen | SunGen w/ AttrPrompt | ProGen | ProGen w/ AttrPrompt | ReGen | ReGen w/ AttrPrompt |
|---|---|---|---|---|---|---|---|---|---|
| NYT | Acc. | 76.11 | 82.05 | 75.82 | 81.65 | 77.05 | 80.93 | 70.01 | 82.18 |
|  | F1 | 76.80 | 82.62 | 76.52 | 82.70 | 76.70 | 81.32 | 68.14 | 82.50 |
| Amazon | Acc. | 58.17 | 66.76 | 54.30 | 63.89 | 58.40 | 66.43 | 34.70 | 58.40 |
|  | F1 | 56.06 | 66.33 | 53.50 | 63.76 | 56.95 | 66.02 | 30.93 | 56.00 |

Table 10: Multi-label classification performance on the arXiv dataset.

| Method | Macro F1 | Micro F1 | Precision@1 | Precision@5 | NDCG@5 | MRR | Price/1k |
|---|---|---|---|---|---|---|---|
| Gold | 27.34 | 58.22 | 73.71 | 27.00 | 79.70 | 82.16 | — |
| SimPrompt | 21.03 | 26.75 | 37.00 | 15.08 | 42.49 | 49.60 | 1.41 |
| AttrPrompt | 27.10 | 37.88 | 49.27 | 18.79 | 54.74 | 61.23 | 1.53 |

## 5.6 Extension to multi-label classification

In this section, we take the first attempt to extend the paradigm of using the LLM as a training data generator to the more challenging multi-label classification setting. In particular, we adopt the arXiv dataset [8] consisting of 98 fine-grained classes, on which we apply both SimPrompt and AttrPrompt. Following [18, 49], we consider different evaluation metrics including Micro/Macro-F1, Precision@$k$, nDCG@$k$, and MRR. The experimental details are in Appendix B. We present the results in Table 10. Similar to our findings for single-label classification, AttrPrompt largely outperforms SimPrompt across all the metrics, which not only strengthens the superiority of AttrPrompt but also opens the door to using LLM as a training data generator for multi-label classification for future work.

## 6 Conclusion

We delve into the realm of training data generation using complex, attributed prompts, which possess the potential to produce a wide range of diverse and attributed generated data. Specifically, we focus on datasets characterized by diverse domains and high cardinality and class-imbalance, and our results demonstrate the superior performance of attributed prompts compared to simple class-conditional prompts. Furthermore, we present a comprehensive empirical study on training data generation that covers essential aspects such as bias, diversity, and efficiency.

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

# A Limitation, Social Impact, and Future Work

We acknowledge that AttrPrompt, while being effective on several tasks we studied, is still far from being perfect. Here we give a honest discussion on the limitation, societal impacts, as well as potential avenues for future works.

## A.1 Limitation

One limitation of this research is that the investigation focuses primarily on text classification. While this provides valuable insights into the performance of attributed prompts compared to simple class-conditional prompts in these specific contexts, the generalizability of the findings to other data types and tasks remains to be explored.

Besides, the proposed approach assumes the label name of the target task following prior works [31, 54] and requires some extent of human efforts in creating the data attributes, as it is still challenging to automatically determine the optimal number/property of attributes without any labeled data and task-specific information. Human-AI collaboration, while beneficial, may inevitably introduce certain human biases into the model [43, 5]. We would like to clarify that in this work, the human raters are blinded to the information for task-specific data during attribute selection. To further mitigate the bias, it is crucial to involve multiple team raters, cross-validation, and additional review to achieve a balanced and unbiased selection of attribute dimensions that contribute positively to the dataset's quality and diversity.

In addition, one issue with LLM-based training data generation is the phenomenon of *hallucination*, wherein the model generates information that may sound plausible but is, in fact, not accurate or grounded in reality. This can lead to the propagation of misinformation, particularly in sensitive areas such as news generation where accuracy is of utmost importance. To alleviate this issue, it is possible to leverage additional fact-checking mechanisms to cross-verify the generated text with a reliable knowledge base or dataset. Furthermore, incorporating an additional layer of human review can also help in mitigating hallucinations and ensuring the faithfulness of generated outputs.

## A.2 Societal impact

The findings of this research have significant implications for the development of trustworthy machine learning models. By addressing the limitations of simple class-conditional prompts and exploring the potential of attributed prompts, the research aims to mitigate biases and enhance the diversity of generated training data. This has a positive social impact as it contributes to the development of more robust and fair machine learning models, reducing the potential for biased decision-making systems and promoting inclusivity in various domains.

During the attribute selection process, we did not use any task-specific data to avoid data feature leakage and minimize human bias. Other than this, we do not involve human subjects research and do not contain any personally identifiable information. Possible misuse may lead to negative outcomes, such as intentionally introducing biases to the generated dataset via attributes.

## A.3 Future work

**Broader Coverage of Task and Modalities.** In future work, it would be valuable to extend the investigation to other types of classification tasks beyond text classification. Exploring the effectiveness of attributed prompts in image classification, audio classification, or other modalities could provide insights into the broader applicability and effectiveness of AttrPrompt.

**Automatic Prompt Optimization.** In this work, we aim to demonstrate the usefulness of attributes and why they are useful through the lens of diversity and bias, and we believe this opens the door for future work to further explore the methodology for automatic attribute discovery and selection for optimal performance, such as exploring automated or semi-automated methods for identifying high-quality attribute dimensions and values. One promising direction is to incorporate a few labeled examples as demonstrations for generating better training data. Under this scenario, leveraging Language Models can aid in automatically detecting diverse attributes, modeling the high-order interactions among attributes as well as estimating attribute value distributions for each class, which further reduce the human efforts for selecting key attributes.

## B  Dataset Preprocessing

### B.1  Multi-class Classification

For Reddit and StackOverflow, we select the classes with more than 65 examples from the original corpus as the target set of topics. For each dataset, we use 50 examples per class for the test set and no more than 10 examples for the validation set (10 for NYT/Amazon and 5 for Reddit/StackOverflow). The remaining data is used to compose the gold training set. It is worth noting that, some of the class names on Reddit may contain toxic information. To eliminate their effects, we filter our label names with Detoxify [19], a tool that leverages the multilingual XLM-RoBERTa [9] for toxic comment identification. We follow [14] to use a threshold of 0.1 to filter out potentially toxic topic classes.

### B.2  Multi-label Classification

For the arXiv multi-label classification, we use the arXiv labels from the website[6] as the label names. We select topics under the category of *Physics*, *Mathematics*, *Computer Science*, *Quantitative Biology*, and *Quantitative Finance*. Besides, we remove the topic of *General Physics*, *General Mathematics*, *General Literature*, *Other Computer Science*, *Other Quantitative Biology* as they only contain generic topics without concrete semantics. In total, there are 98 classes in general.

To generate examples using the SimPrompt approach, we follow a specific procedure. For each document, we begin by sampling from the distribution of label numbers, utilizing prior knowledge obtained from the original training set. This enables us to determine the precise number of labels, denoted as $n$, that should be associated with the document. Once we have determined $n$, we proceed to randomly select $n$ classes from the original set of 98 labels. These chosen classes are then utilized as the input for prompting the LLM, forming a foundation for generating the examples.

To generate examples using the AttrPrompt approach, we employ a distinct methodology. For each document, we consider a set of attributes, namely `subtopics`, `technique`, `writing style`, and `length`. It is important to note that, unlike multi-class classification, where an additional step is typically taken to filter out irrelevant subtopics, our multi-label classification process follows a different approach. In our method, we introduce an additional step called *merging* to handle the different subtopics.

To determine the merging of subtopics, we utilize the fuzzy matching score[7], which calculates the similarity between the subtopics. Setting the threshold at $\gamma = 90$, we proceed to merge subtopics whenever their similarity score exceeds this threshold. Consequently, specific subtopics may be associated with multiple classes, providing a more comprehensive representation of the document content. When generating text utilizing these subtopics, the corresponding labels are determined by the classes associated with the specific subtopics. This ensures that the generated examples accurately reflect the relevant classes based on the subtopic associations. The prompt format is deferred in section H.

## C  Implementation Details

### C.1  Hardware information

All experiments are conducted on *CPU*: Intel(R) Core(TM) i7-5930K CPU @ 3.50GHz and *GPU*: NVIDIA GeForce RTX A5000 GPUs using python 3.8, Huggingface 4.6.0 and Pytorch 1.10.

### C.2  Parameter configuration

We keep the parameter $\text{top\_p} = 1.0$ and temperature $t = 1.0$ for calling ChatGPT APIs [38] for the training data generation part. For finetuning the classifier, we optimize the model using AdamW [30] with a linear warmup of the first 5% steps and linear learning rate decay. The maximum number of tokens per sequence is 128. Table 11 lists the hyperparameters used for AttrPrompt and SimPrompt. For the generated synthetic dataset, we stick to the strict zero-shot learning setting [31], train all

---

[6] https://arxiv.org/
[7] https://github.com/seatgeek/thefuzz

the models for 6 epochs and use the model from the last epoch *without using the validation set* for evaluation. For the original clean dataset, we train all models for 6 epochs and use the model with the best performance on the validation set for evaluation.

Table 11: Hyperparameters for fine-tuning classifiers on different tasks.

| Backbone | Learning Rate lr | Batch Size | Training Epochs $E$ | Weight Decay | Warmup Ratio |
|---|---|---|---|---|---|
| BERT-base-uncased [12] | 5e-5 | 32 | 6 | 1e-4 | 6% |
| TinyBERT [23] | 1e-4 | 32 | 6 | 1e-4 | 6% |
| DistilBERT-base-uncased [46] | 5e-5 | 32 | 6 | 1e-4 | 6% |
| DeBERTa-V3-base [21] | 5e-5 | 32 | 6 | 1e-4 | 6% |
| DeBERTa-V3-large [21] | 2e-5 | 32 | 6 | 1e-4 | 6% |

Table 12: The reference of model checkpoints used in this study.

| Model | Link |
|---|---|
| BERT-base-uncased [12] | https://huggingface.co/bert-base-uncased |
| TinyBERT [23] | https://huggingface.co/huawei-noah/TinyBERT_General_4L_312D |
| DistilBERT-base-uncased [46] | https://huggingface.co/distilbert-base-uncased |
| DeBERTa-v3-base [21] | https://huggingface.co/microsoft/deberta-v3-base |
| DeBERTae-v3-large [21] | https://huggingface.co/microsoft/deberta-v3-large |
| SentenceBERT [44] | https://huggingface.co/sentence-transformers/all-mpnet-base-v2 |
| SimCSE [16] | https://huggingface.co/princeton-nlp/sup-simcse-bert-base-uncased |
| COCO-DR [56] | https://huggingface.co/OpenMatch/cocodr-base-msmarco |

## C.3 Model checkpoint

All the checkpoint used in fine-tuning and linear probing experiments are listed in table 12.

# D Additional Experimental Results

## D.1 Linear probing with generated datasets

In this section, we evaluate the generated dataset under the linear probing setting, where a linear classifier is trained over the frozen pretrained features. To make a comprehensive evaluation, we employ four text embedding methods, namely BERT-CLS [12], SentenceBERT [44], SimCSE [16] and COCO-DR [56], and present the results in Table 13. We can see that AttrPrompt consistently outperforms SimPrompt, MetaPrompt and surpasses the Gold in the NYT dataset. This further indicates the datasets generated with AttrPrompt has higher quality than two baselines and this superiority is robust to the model choices.

## D.2 Additional results on attribute selection

Table 14 shows the result of AttrPrompt without human selection of attributes on two datasets. While a slight performance drop is observed, the result of AttrPrompt without attribute selection still significantly outperform the SimPrompt. This demonstrates the robustness of AttrPrompt performance to potentially irrelevant attributes.

## D.3 Additional results on data/budget efficiency

In Section 5.3, we discuss the data/budget efficiency of AttrPrompt and SimPrompt on NYT and Amazon datasets; here, we provide additional results on the remaining datasets in Figure 8. We draw conclusions similar to what is stated in Section 5.3 that AttrPrompt exhibits better data/budget efficiency than SimPrompt.

## D.4 Different temperature parameters for ChatGPT

Temperature ($t$) is one crucial hyperparameter of LLMs that controls the diversity of the generated text [20, 10], while the studied attributed prompts are also for diversifying the generated data. We are then curious about the effectiveness of the temperature and how it compares to the AttrPrompt. We

Table 13: Performance of linear probing with different pre-trained encoders.

| Sequence Encoder | Method | NYT | | Amazon | | Reddit | | StackExchange | |
|---|---|---|---|---|---|---|---|---|---|
| | | Acc | F1 | Acc | F1 | Acc | F1 | Acc | F1 |
| BERT-CLS [12] | Gold | 65.39 | 70.26 | 66.73 | 68.76 | 55.73 | 58.00 | 17.17 | 17.68 |
| | SimPrompt | 60.69 | 62.26 | 37.95 | 38.07 | 27.41 | 27.60 | 12.27 | 13.71 |
| | MetaPrompt | 58.43 | 57.05 | 27.52 | 24.43 | 23.93 | 23.99 | 11.40 | 12.60 |
| | AttrPrompt | 62.38 | 64.17 | 48.18 | 49.46 | 45.03 | 45.95 | 17.33 | 17.64 |
| SentenceBERT [44] | Gold | 81.16 | 84.43 | 80.95 | 83.30 | 80.76 | 81.37 | 55.13 | 54.25 |
| | SimPrompt | 84.32 | 85.91 | 67.55 | 68.29 | 69.94 | 70.62 | 45.37 | 47.79 |
| | MetaPrompt | 87.65 | 87.11 | 67.16 | 65.78 | 79.26 | 69.15 | 45.71 | 47.54 |
| | AttrPrompt | 88.77 | 88.36 | 68.10 | 69.11 | 71.62 | 71.73 | 46.80 | 48.35 |
| SimCSE [16] | Gold | 78.02 | 80.17 | 73.20 | 75.13 | 73.75 | 74.26 | 45.26 | 41.86 |
| | SimPrompt | 78.93 | 78.86 | 52.40 | 52.83 | 54.78 | 53.86 | 30.98 | 32.92 |
| | MetaPrompt | 78.17 | 78.27 | 53.53 | 52.55 | 51.30 | 48.93 | 30.54 | 31.50 |
| | AttrPrompt | 81.41 | 81.04 | 60.66 | 61.32 | 59.22 | 59.15 | 30.31 | 31.43 |
| COCO-DR [56] | Gold | 80.66 | 82.60 | 76.13 | 78.34 | 79.63 | 80.04 | 61.02 | 63.77 |
| | SimPrompt | 84.37 | 84.17 | 55.66 | 55.28 | 63.66 | 64.71 | 43.22 | 41.50 |
| | MetaPrompt | 84.51 | 84.26 | 57.96 | 56.58 | 65.36 | 65.91 | 44.30 | 42.97 |
| | AttrPrompt | 86.74 | 86.43 | 60.47 | 62.03 | 66.73 | 66.71 | 45.44 | 43.76 |
| Average | Gold | 76.31 | 79.37 | 74.25 | 76.38 | 72.47 | 73.42 | 44.65 | 44.39 |
| | SimPrompt | 77.08 | 77.80 | 53.39 | 53.62 | 53.95 | 54.20 | 32.96 | 33.98 |
| | MetaPrompt | 77.19 | 76.67 | 51.54 | 49.84 | 54.96 | 52.00 | 32.99 | 33.65 |
| | AttrPrompt | 79.83 | 80.00 | 59.35 | 60.48 | 60.65 | 60.89 | 34.97 | 35.30 |

| | NYT | Amazon |
|---|---|---|
| GPT-3.5 Zero-shot | 69.84 | 54.56 |
| SimPrompt | 76.34 | 56.96 |
| AttrPrompt | 82.26 | 65.87 |
| AttrPrompt w/o selection | 81.08 | 63.76 |

Table 14: The Performance of AttrPrompt with/without attribute selection.

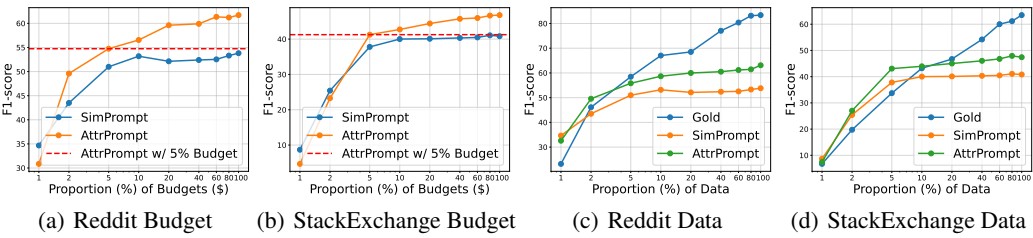

(a) Reddit Budget    (b) StackExchange Budget    (c) Reddit Data    (d) StackExchange Data

Figure 8: The comparisons on budget efficiency and data efficiency on Reddit and StackExchange.

study different values of the temperature using the NYT dataset and present the results in Table 15. From the results, we can see that compared with the temperature, AttrPrompt brings more significant performance gain, demonstrating its superiority over temperature tuning.

Table 15: Study of the temperature.

| Method | $t = 1.0$ | | $t = 1.5$ | | $t = 2.0$ | |
|---|---|---|---|---|---|---|
| | Acc. | F1 | Acc. | F1 | Acc. | F1 |
| SimPrompt | 76.00 | 76.34 | 76.78 | 77.31 | 76.55 | 77.42 |
| AttrPrompt | 81.30 | 82.26 | 81.47 | 82.23 | 79.47 | 79.86 |

## D.5 The performance with respect to long-tail classes

As we have seen in Table 2, the original training sets of the involved datasets have severe long-tail classes issue since the imbalance ratio is high, yet the generated dataset are class-balanced, we are then curious how the class balance in the generated dataset benefits the model performance on long-tail classes. We take the NYT dataset as an example and plot the per-class F1 score of Gold, SimPrompt,

and AttrPrompt in Figure 9, where the x-axis is classes sorted by their number of data in the Gold dataset in descending order. From the figure, we can see that out of 26 classes, AttrPrompt renders the best per-class F1 score on 10 classes, which is 13 for Gold and 3 for SimPrompt. Notably, for classes with few examples in the Gold set (the rightmost 4 classes in the figure), AttrPrompt is better than the Gold and SimPrompt, especially for the class "*abortion*" with the fewest examples. This suggests a data-centric way to handle the long-tail class issue in topic classification: one may use LLMs to generate class-balanced training set or augment the existing training set with the LLM-generated data such that the augmented dataset is class-balanced, the in-depth study of which is left as future work.

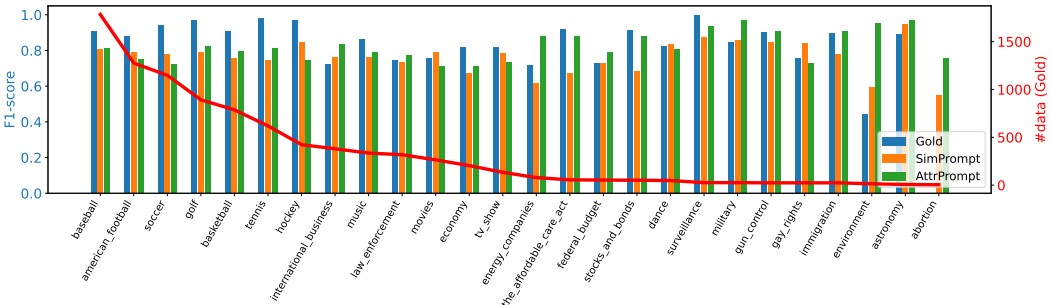

Figure 9: Per-class F1-score of the NYT dataset.

## D.6 Experiment results on other datasets

To provide a comprehensive evaluation of AttrPrompt, we conducted additional empirical studies using datasets commonly employed in prior works, which typically exhibit *low cardinality*. Consistent with [55, 54, 57], we focused on four specific tasks: IMDB, SST-2, Yelp, and AG News. The IMDB, SST-2, and Yelp datasets involve binary sentiment classification, while the AG News dataset entails news classification with only four classes. This selection allows for diverse evaluation across different classification scenarios.

To ensure a fair comparison, we utilized a smaller pre-trained language model (PLM) called Distill-BERT [46] as the classifier, in line with [55, 54, 15]. For the purpose of fine-tuning the classifier, we generated a total of 6000 examples, evenly distributed among the classes. For fine-tuning, we use a learning rate of $2 \times 10^{-5}$ with 5 epochs and use the model after the final epoch for evaluation.

The results of our evaluation are presented in table 16. Overall, we observe that AttrPrompt demonstrates competitive performance across all four tasks. It outperforms our direct baseline (SimPrompt) by an impressive margin of 3.8%. Moreover, when compared to other baselines incorporating noisy-aware learning techniques, AttrPrompt still achieves promising performance, despite using the standard cross-entropy loss. This suggests that in addition to leveraging noisy-robust learning objectives, enhancing the diversity of input prompts serves as an effective alternative to improving downstream performance. It is interesting yet important to combine these different learning paradigms together to further boost the performance.

## E  Regional Bias with Manual Annotations

In the main body of the paper, we study the regional bias in the NYT dataset via an attribute classifier trained with the location attribute of the prompt associated with each generated data. Here, we manually annotate location for 100 samples from each of the Gold, SimPrompt, and AttrPrompt dataset, and compare manual annotations with the prediction of the attribute classifier in Table 17 (the first three rows). Note that we include "*unkown*" as an option in manual annotation to absorb text without clear location specification. From the results, we can see that the attribute classifier largely aligns with manual annotations, since the accuracy is greater than 0.95 on the Gold and AttrPrompt and 0.82 on SimPrompt; and the result on SimPrompt is lower since SimPrompt tend to generate more unknown location data.

With manual annotations, we also check how well ChatGPT can follow the required location attribute in the prompts. We compute the accuracy between the location attribute in the prompt associated with

Table 16: Results with recent baselines using DistilBERT [46]. *: Results are copied from the previous papers [55, 54, 15, 57].

| Method/Dataset | IMDB | SST-2 | Yelp | AG News | Avg. |
|---|---|---|---|---|---|
| *Simple fine-tuning with standard cross-entropy loss* | | | | | |
| SimPrompt | 85.64 | 79.38 | 88.39 | 74.30 | 81.93 |
| AttrPrompt | 85.92 | 83.64 | 90.60 | 82.74 | 85.73 |
| *Other results with dedicated training techniques (for reference only)* | | | | | |
| Zero-shot Prompting* | 77.31 | 82.63 | 78.10 | 73.20 | 77.81 |
| ZeroGen* | 82.77 | 80.41 | 87.84 | 76.48 | 81.88 |
| SuperGen* | 85.05 | 84.28 | 89.88 | 77.40 | 84.15 |
| ProGen* | 87.20 | 84.12 | 89.39 | 80.81 | 85.38 |
| SunGen* | 89.45 | 89.38 | 89.19 | 80.49 | 87.12 |
| ReGen* | 85.32 | 87.84 | 89.00 | 80.18 | 85.58 |

each data and that annotated by human on the 100 AttrPrompt samples (the last row of Table 17), which is 0.96. This means that ChatGPT is able to faithfully generate news with desired location specifications.

Table 17: Accuracies based on manual annotation of the location attribute of the NYT dataset.

| Comparison | Data Source | Accuracy |
|---|---|---|
| | Gold | 0.96 |
| Attribute Classifier v.s. Human | SimPrompt | 0.82 |
| | AttrPrompt | 0.95 |
| Attribute in Prompt v.s. Human | AttrPrompt | 0.96 |

We also plot the pie charts using manual annotations of location in Figure 10, and obtain similar findings as in Section 4.2, *i.e.*, the data in Gold and SimPrompt are largely biased towards "*North Amarica*", while that of AttrPrompt is relatively balanced.

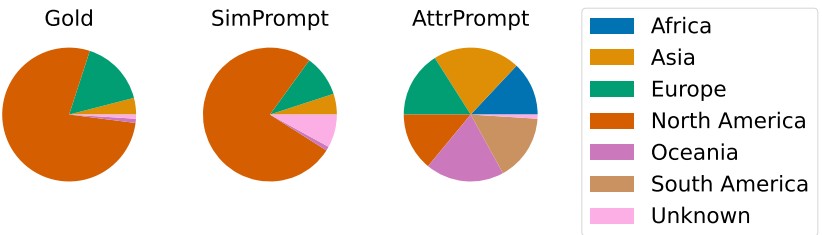

Figure 10: Pie charts of the "*location*" attribute based on manual annotations.

# F  Additional Bias Analysis

## F.1  Biases other than regional bias

In addition to the regional bias we study in the main body of the paper, here we investigate bias in other datasets (Figure 11). From the results, we can see that the dataset generated with Gold and SimPrompt exhibit different level of biases, which shows that in general the biases exists in both generated and real dataset, while dataset generated with AttrPrompt has relatively balanced attribute distribution.

## F.2  Bias analysis in AttrPrompt

Here, we compare the distribution of predicted "*location*" in the dataset generated with AttrPrompt to that with the Gold and SimPrompt in Figure 12. We can see that AttrPrompt renders a relatively balanced "*location*" distribution.

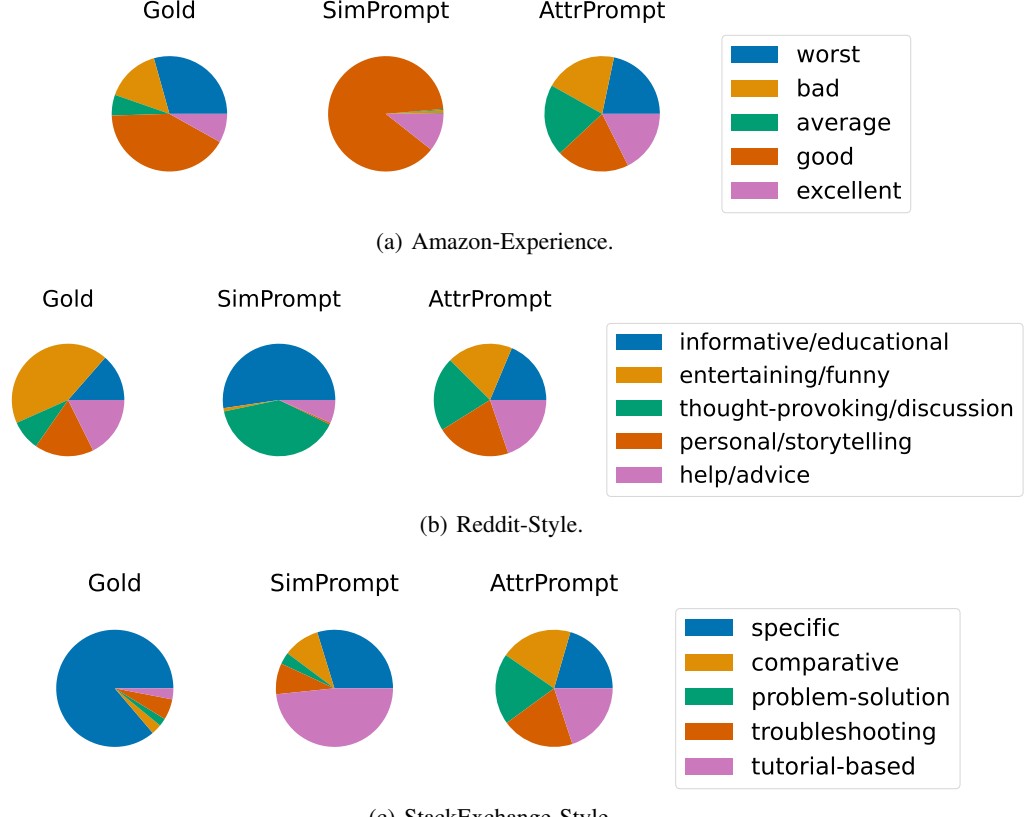

(a) Amazon-Experience.

(b) Reddit-Style.

(c) StackExchange-Style.

Figure 11: Pie charts of predicted attribute distributions on different datasets.

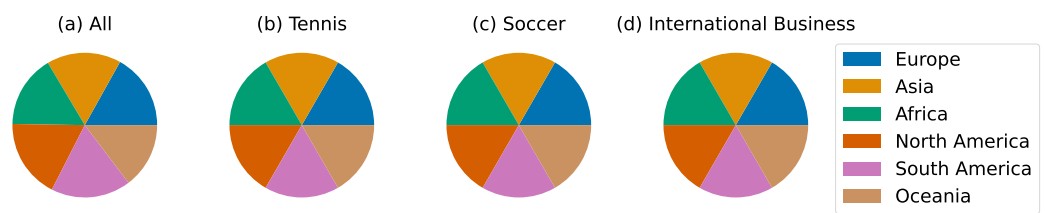

Figure 12: Pie charts of the distributions of "*location*" predicted by an attribute classifier for the NYT AttrPrompt dataset. (a) is "*location*" distribution over the whole dataset, while others are specific classes.

# G   The Prompt Format for LLM-Generated Attributes

## G.1   Prompts for NYT Dataset

### G.1.1   Key Attributes

**Prompts to ChatGPT.** What do you think are important attributes to generate some diverse news from NYT under a specific topic, such as Football, Federal Budget. Examples: subtopics, writing style...

**Response from ChatGPT.** Important attributes of news from the New York Times under a specific topic could include:

Subtopics: exploring different angles and subtopics within the main topic can lead to a wider range of news stories.

Writing style: varying the writing style can help to make the news stories more interesting and engaging for readers. This can include using different tones, perspectives, and sentence structures.

Sources: using a variety of sources, such as experts, eyewitnesses, and affected individuals, can provide different perspectives and add diversity to the news stories.

Geographical locations: covering the topic from different geographical locations can help to show how it affects different communities and provide a global perspective.

Timeliness: covering current events and breaking news related to the topic can add diversity and relevance to the news stories.

Multimedia: using a range of multimedia elements, such as images, videos, and infographics, can help to add variety and appeal to the news stories.

Length: varying the length of the news stories, from short and concise to longer and more in-depth, can help to provide a wider range of information and cater to different readers' preferences.

**Selected Attributes**

1. Subtopics
2. Writing Styles
3. Geographical locations (determined by six continents by default)
4. Length (determined by {30-80} and {100-150} by default)

### G.1.2   Attributes Details

**Prompts to ChatGPT for Similar Classes.**   List 5 similar classes for {class-name} news on NYT. The set of classes is listed as: {[a list of class-names]}.

**Prompts to ChatGPT for Subtopics.**   List 10 diverse subtopics for {class_name} news on NYT.

**Response from ChatGPT.**   See section H for details.

### G.1.3   Writing Styles

**Prompts to ChatGPT.**   List 5 different writing styles of NYT news.

**Response from ChatGPT.**

1. Investigative reporting
2. Opinion/editorial
3. Human interest/profiles
4. Long-form narrative
5. Breaking news

### G.2 Prompts for Amazon Dataset

#### G.2.1 Key Attributes

**Prompts to ChatGPT.** What do you think are important attributes to generate some diverse product reviews. Examples: writing style...

**Response from ChatGPT.** When writing a review for a product, here are five important attributes to consider:

Product Type: Clearly mention the type of product you are reviewing, such as a smartphone, laptop, or fitness tracker. This helps readers understand the category and purpose of the product.

Brand: Specify the brand of the product as it often influences quality, reputation, and customer support. Discuss the brand's overall credibility and whether it aligns with your expectations.

User Experience: Evaluate the overall user experience of the product. Discuss its speed, accuracy, reliability, and efficiency in performing its intended tasks. Highlight any exceptional or lacking performance aspects.

Quality and Durability: Assess the quality of the product, including the materials used, construction, and overall durability. Discuss whether it feels well-made, solid, and likely to withstand regular use over time.

Features and Functionality: Describe the specific features and functions of the product. Highlight any unique or standout features that enhance its usability or set it apart from similar products in the market.

#### G.2.2 Selected Attributes

1. Product Name
2. Brands
3. User Experience
4. Writing Styles
5. Length

#### G.2.3 Attributes Details

**Prompts to ChatGPT for Similar Classes.** List 5 similar classes for {product-class} products on Amazon. The set of classes is listed as: {[a list of class-names]}.

**Prompts to ChatGPT for Product Name/Brands.** List 20 diverse product names/brands for {product-class} on Amazon.

**Response from ChatGPT.** See section H.2 for details.

#### G.2.4 Usage Experience

**Prompts to ChatGPT.** List 5 different usage experiences of Amazon reviews.

**Response from ChatGPT.**

1. Worst: A highly unsatisfactory or problematic experience. Users may encounter multiple issues, such as usability problems, errors, and frustrations.

2. Bad: A below-average experience. Users may encounter some difficulties or frustrations while interacting with the product or service, but the overall experience falls short of meeting their expectations.

3. Average: A neutral or middling experience. Users may not encounter major issues, but the experience lacks any exceptional aspects or standout features.

4. Good: A positive and satisfactory experience. Users have a smooth and enjoyable experience with minimal issues or obstacles.

5. Excellent: The highest level of satisfaction. Users have a seamless, delightful, and memorable experience.

### G.2.5 Writing Styles

**Prompts to ChatGPT.** List 5 different writing styles of Amazon reviews.

**Response from ChatGPT.**

1. Detailed Review: A detailed review provides a comprehensive analysis of the product, covering various aspects such as features, performance, durability, and user experience.

2. Comparative Review: A comparative review compares the product with similar options available on Amazon.

3. Pros and Cons Review: A pros and cons review presents a balanced assessment of the product by listing its positive and negative aspects.

4. Personal Experience Review: A personal experience review shares the reviewer's firsthand experience with the product.

5. Recommendation Review: A recommendation review offers a clear recommendation or endorsement of the product.

### G.2.6 Prompt Template

See table 18 for details.

Table 18: Prompt template for the Amazon Product Review dataset.

| Method | Prompt |
|---|---|
| SimPrompt | Suppose you are a review writer. Please write a review for {product-class} in Amazon. |
| AttrPrompt | Suppose you are a review writer. Please write a review for {product-class} product in Amazon following the requirements below:
1. The review should be about the product of {subtopic};
2. The brand for the product should be {brand};
3. Should be in length between {length:min-words} and {length:max-words} words;
4. Should describe the usage experience {usage-experience}
5. The writing style of the review should be {style};
6. the review must be relevant to {product-class} and irrelevant to: {similar-class}. |

### G.3 Prompts for Reddit Dataset

### G.3.1 Key Attributes

**Prompts to ChatGPT.** List 5 important attributes to generate some diverse reddit posts under a specific topic, like build_a_pc, premed, etc. Examples: subtopics, keywords, writing style...

**Response from ChatGPT.** Informative Guides and Tutorials: Create posts that provide step-by-step guides, tutorials, or informative content. These can include detailed instructions, tips, or recommendations to help readers in their journey. For example, in "build_a_pc," you could create a guide on assembling a gaming PC or optimizing system performance.

Writing Style: Utilize different writing styles to create diverse posts. For instance, you can write informative guides or tutorials, opinion pieces, comparison reviews, troubleshooting tips, or personal stories related to the topic.

Discussion Starters: Pose thought-provoking questions or statements to initiate discussions and engage the Reddit community. These can be open-ended queries, asking for opinions or experiences, or presenting a controversial viewpoint to encourage debate and dialogue.

Personal Experiences: Share personal stories or experiences related to the topic. This allows you to connect with the community on a more personal level and provide insights based on your firsthand experiences. It can include successes, challenges, lessons learned, or interesting anecdotes.

Table 19: Prompt template for the Reddit dataset.

| Method | Prompt |
|---|---|
| SimPrompt | Suppose you are a writer for posts in Reddit. Please give a post for {post-class} in Reddit. |
| AttrPrompt | Suppose you are a review writer. Please give a post in Reddit on {post-class} community following the requirements below:
1. The post should focus on {experience};
2. Should be in length between {length:min-words} and {length:max-words} words;
3. The writing style of the post should be {style};
4. Should mention the resource of {resource};
5. the post must be relevant to {post-class} and irrelevant to: {similar-class}. |

Resources and Recommendations: Provide valuable resources such as articles, websites, tools, or recommended products/services related to the topic. These resources can help the community access additional information, useful tools, or trusted recommendations, enhancing their understanding and engagement with the subject.

### G.3.2  Selected Attributes

1. Personal Experiences
2. Resources
3. Writing Styles
4. Length

### G.3.3  Attributes Details

**Prompts to ChatGPT for Similar Classes.**   List 5 similar classes for {post-class} products on Reddit. The set of classes is listed as: {[a list of class-names]}.

**Prompts to ChatGPT for Personal Experiences/Resources.**   List 20 diverse Personal Experiences/Resources for posts about {post-class} in Reddit.

**Response from ChatGPT.**   See section H.3 for details.

### G.3.4  Writing Styles

**Prompts to ChatGPT.**   List 5 different writing styles of Reddit Posts.

**Response from ChatGPT.**

1. Informative/Educational: Create posts that aim to provide valuable information or educate the community on a particular topic. Share informative articles, research findings, guides, tutorials, or explain complex concepts in a simplified manner.

2. Entertaining/Funny: Craft posts that are meant to entertain or bring humor to the community. Share funny stories, jokes, memes, amusing videos, or any form of content that can elicit laughter and enjoyment.

3. Thought-Provoking/Discussion: Initiate posts that spark thoughtful discussions and invite users to share their opinions or experiences. Pose open-ended questions, present philosophical dilemmas, discuss current events, or explore societal issues to foster engaging conversations.

4. Personal/Storytelling: Share personal experiences, anecdotes, or narratives that others can relate to. Tell compelling stories, share achievements, challenges, or life lessons that can inspire or resonate with the community.

5. Help/Advice: Create posts where you seek help or offer advice to the community. Ask for guidance on a specific problem, seek recommendations, provide support, or share expertise in a particular field.

### G.3.5  Prompt Template

See table 19 for details.

### G.4 Prompts for StackExchange Dataset

### G.4.1 Key Attributes

**Prompts to ChatGPT.** What do you think are important attributes to generate diverse posts under a specific topic on the stackexchange forum, like 3D meshes, server, etc. For example, one important attribute could be writing style...

**Response from ChatGPT.** When generating diverse posts under a specific topic on the Stack Exchange forum, several important attributes can contribute to the variety and richness of the discussions:

Depth of Knowledge: Participants with different levels of expertise can offer insights and perspectives from beginner to advanced, allowing for a broader range of questions and answers.

Diverse Perspectives: Encouraging participants from various backgrounds, industries, and experiences can bring fresh viewpoints and approaches to the topic.

Objectives: Encouraging participants to state their objectives or goals in their posts can help generate diverse content. Some posts may aim to seek advice, share experiences, provide step-by-step guides, analyze specific problems, or discuss broader concepts.

Problem Scenarios: Encourage users to discuss real-world problem scenarios related to the topic. This helps generate diverse posts by exploring different use cases, challenges, and approaches to problem-solving.

Problem Solving: Promoting problem-solving skills and encouraging users to share their challenges, roadblocks, and solutions can foster a collaborative and dynamic environment.

Clear Communication: Emphasizing clear and concise communication helps to ensure that questions are well-articulated and that answers are easily understood by a wide range of users.

### G.4.2 Selected Attributes

1. Problem Scenarios
2. Depth of Knowledge
3. Writing Styles
4. Length

**Prompts to ChatGPT for Problem Scenarios.**  List 20 diverse Problem Scenarios for posts about {post-class} in StackExchange.

**Response from ChatGPT.**  See section H.4 for details.

### G.4.3 Writing Styles

**Prompts to ChatGPT.**  List 5 different writing styles of StackExchange Posts.

**Response from ChatGPT.**

1. Specific: Be specific in your question by including relevant details, such as error messages, code snippets, or specific scenarios, to help the community provide accurate and targeted answers.
2. Comparative: Compare different approaches, technologies, or methodologies, and ask for insights or opinions from the community on their strengths, weaknesses, and suitability for your specific situation.
3. Problem-Solution: Present a problem or challenge you are encountering and outline any attempted solutions, seeking feedback or alternative approaches from the community to help you find an optimal solution.
4. Troubleshooting: Describe a specific issue or error you are encountering and provide any relevant error messages or logs, seeking assistance in identifying the cause and potential solutions.

Table 20: Prompt template for the StackExchange dataset.

| Method | Prompt |
|--------|--------|
| SimPrompt | Suppose you are a writer for posts in StackExchange. Please give an example of a question post in StackExchange about {post-class}. |
| AttrPrompt | Please give an example of a question post in StackExchange about {post-class} following the requirements below:
1. Should focus on the scenario of {scenario};
2. Should be in length between {length:min-words} and {length:max-words} words;
3. The writing style of the question should be {style};
4. The question should be in {depth};
5. the review must be relevant to {post-class} and irrelevant to: {similar-class}. |

5. Tutorial-Based: Ask for guidance or step-by-step instructions on how to achieve a specific task or goal, seeking expertise and advice from the community.

### G.4.4 Depth of Knowledge

**Prompts to ChatGPT.** List 5 different levels of depth of knowledge for StackExchange Posts.

**Response from ChatGPT.**

1. Beginner Level: Questions that address basic concepts and seek introductory explanations or guidance. These questions are suitable for individuals who are new to the topic.

2. Intermediate Level: Questions that require a moderate understanding of the topic and seek more specific information or solutions to intermediate-level problems.

3. Advanced Level: Questions that delve into complex concepts or advanced techniques, requiring a deeper understanding of the topic and seeking advanced solutions or insights.

4. Expert Level: Questions that involve specialized or advanced knowledge within a specific field, targeting experts or professionals seeking detailed and intricate solutions.

### G.4.5 Prompt Template

See table 21 for details.

### G.5 Prompts for arXiv Dataset

### G.5.1 Key Attributes

**Prompts to ChatGPT.** What do you think are important attributes to generate diverse papers under a specific topic on arXiv. For example, one important attribute could be writing style...

**Response from ChatGPT.** Generating diverse papers under a specific topic on platforms like arXiv involves considering various important attributes. Here are a few attributes that can contribute to generating diverse papers:

Writing Style: Encouraging papers with different writing styles, such as technical, expository, theoretical, or empirical, can bring diversity to the presentation and appeal to a wider range of readers.

Subtopics: Promoting papers that explore different subtopics within the broader topic can provide comprehensive coverage and delve into specific areas of interest.

Techniques: Encouraging papers that employ different research methodologies, such as experimental, computational, or analytical, can bring diverse approaches to studying the topic.

Data Sources: Promoting papers that utilize diverse data sources, such as surveys, simulations, real-world datasets, or case studies, can offer different perspectives and insights into the topic.

Interdisciplinary Perspectives: Encouraging papers that incorporate interdisciplinary perspectives, drawing insights from multiple fields or combining methodologies from different disciplines, can contribute to a richer understanding of the topic.

### G.5.2 Selected Attributes

1. Subtopics
2. Techniques

Table 21: Prompt template for the arXiv multi-label dataset.

| Method | Prompt |
|---|---|
| SimPrompt | Please give a synthetic paper abstract about {paper-classes}. |
| AttrPrompt | Please write an abstract of a {paper-classes} paper in arXiv following the requirements below:
1. Should focus on the {subtopic};
2. Should be in length between {length:min-words} and {length:max-words} words;
3. The paper should use the technique relevant to {technique};
4. The style of the paper should be in {style}; |

3. Writing Styles

4. Length

**Prompts to ChatGPT for Subtopics/Techniques.** List 20 diverse subtopics/techniques for paper about {paper-class} in arXiv.

**Response from ChatGPT.** See section H.5 for details.

### G.5.3 Writing Styles

**Prompts to ChatGPT.** List 5 different writing styles of arXiv papers.

**Response from ChatGPT.**

1. Experimental: This style focuses on presenting empirical research with a strong emphasis on experimental design, data collection, and statistical analysis.

2. Theoretical: This style emphasizes the development of new theoretical frameworks or conceptual models.

3. Review: This style involves synthesizing and analyzing existing literature on a specific topic.

4. Case Study: This style presents an in-depth examination of a specific case or situation, aiming to provide insights into complex phenomena or phenomena that are context-specific.

### G.5.4 Prompt Template

See table 21 for details.

## G.6 Prompts for AG News Dataset

The prompts are the same as the NYT dataset. The only difference is the subtopics for different labels. The subtopic information is listed in H.7 for details.

## G.7 Prompts for Yelp/SST-2/IMDB Dataset

### G.7.1 Key Attributes

**Prompts to ChatGPT.** What do you think are important attributes to generate some diverse reviews with sentiments? E.g. writing styles

**Response from ChatGPT.** Writing Styles: Utilizing different writing styles such as descriptive, narrative, persuasive, or humorous can create a diverse range of tones and emotions in the reviews.

Language and Tone: Varying the language and tone used in the reviews can evoke different sentiments. This can include formal or casual language, positive or negative tone, or even incorporating elements of storytelling.

Personal Experience: Sharing personal experiences and anecdotes in the reviews can add authenticity and emotional depth, allowing readers to connect with the sentiments expressed.

Descriptive Details: Including specific and vivid descriptions of the ambiance, decor, flavors, smells, or textures of the food can create a sensory experience that evokes various sentiments.

Table 22: Prompt template for the Yelp dataset.

| Method | Prompt |
|---|---|
| SimPrompt | Suppose you are a review writer. Please give an example of a {sentiment} review in Yelp. |
| AttrPrompt | Please write a {sentiment} review for a restaurant, following the requirements below:
1. The overall review should be {sentiment};
2. It should be a {descriptive-details} restaurant ;
3. Should focus on {personal-experience};
4. Should be in length between {length:min-words} and {length:max-words} words;
5. The writing style of the review should be {style}; |

## G.7.2 Selected Attributes

1. Personal Experience

2. Descriptive Details

3. Writing Styles

4. Length

## G.7.3 Personal Experiences/Descriptive Details

**Prompts to ChatGPT.** Details are listed in Appendix H.7 and H.8.

## G.7.4 Writing Styles

**Prompts to ChatGPT.** List 5 different writing styles of Restaurant Reviews (for Yelp).

**Response from ChatGPT.**

1. Descriptive Style: Focuses on providing detailed descriptions of the restaurant's ambiance, decor, and overall atmosphere. It aims to paint a vivid picture of the dining experience.

2. Personal Narrative Style: Incorporates personal anecdotes and experiences from the visit to the restaurant. It adds a personal touch to the review and allows the reader to connect with the reviewer's perspective.

3. Analysis Style: Takes a more analytical approach by evaluating various aspects of the restaurant, such as food quality, service, cleanliness, and value for money.

4. Comparative Style: Compares the restaurant to similar establishments in terms of cuisine, price range, service quality, or other relevant factors.

5. Gastronomic Style: Focuses primarily on the food and beverage offerings of the restaurant.

**Prompts to ChatGPT.** List 5 different writing styles of Movie Reviews (for IMDB/SST-2).

**Response from ChatGPT.**

1. Descriptive: Focuses on providing a detailed and vivid description of the movie and its elements.

2. Analytical: Analyzes and evaluates various aspects of the movie, such as the plot, acting, cinematography, and direction.

3. Persuasive: Aims to convince the reader to watch or avoid the movie by presenting strong arguments and opinions.

4. Comparative: Compares the movie to other films within the same genre or by the same director.

## G.7.5 Prompt Template

See table 22 and 23 for details.

Table 23: Prompt template for the SST-2/IMDB dataset.

| Method | Prompt |
|---|---|
| SimPrompt | Suppose you are a review writer. Please give an example of a {sentiment} review for a movie. |
| AttrPrompt | Please write a {sentiment} review for a movie, following the requirements below:
1. The overall review should be {sentiment};
2. It should be a {descriptive-details} movie ;
3. Should focus on {personal-experience};
4. Should be in length between {length:min-words} and {length:max-words} words;
5. The writing style of the review should be {style}; |

# H  Details for LLM-Generated Attributes

## H.1  NYT Attributes

### H.1.1  Subtopics

We randomly select 5 categories in NYT dataset and display the corresponding subtopic attributes for each category:

- astronomy:
    - Discoveries of exoplanets
    - Black holes and their role in shaping galaxies
    - The search for extraterrestrial life
    - Gravitational waves and the study of the universe's origins
    - The use of telescopes to explore the universe
    - The mysteries of dark matter and dark energy
    - Solar flares and their impact on Earth
    - The history of the universe and its evolution over time
    - Exploring the possibility of space tourism
    - The exploration of our neighboring planets, such as Mars and Venus.

- baseball:
    - Recent controversy surrounding sign-stealing scandal in MLB
    - Breakdown of top prospects in minor league baseball
    - Analysis of new rule changes for upcoming baseball season
    - Coverage of recent World Series champions and their success
    - In-depth profile of influential baseball figures, such as managers or players
    - Updates on retired players and their post-baseball careers
    - Highlighting standout performances by individual players or teams in recent games
    - Coverage of international baseball leagues and their top players

- economy:
    - Job market and employment rates
    - Interest rates and monetary policy
    - Inflation and deflation
    - Economic growth and GDP
    - Consumer spending and retail sales
    - Income inequality and poverty
    - GDP growth and contraction
    - Labor market trends
    - Economic impacts of natural disasters and pandemics
    - Housing market and real estate

- federal_budget:
    - Deficit reduction strategies
    - Government spending priorities

- Tax reform proposals
- Agency budget cuts and restructuring
- Budget negotiations and debates
- National debt projections
- Fiscal policy decisions
- Congressional budget proposals
- Infrastructure spending plans
- Public opinion on federal budget issues

- movies:
  - Box office records and trends for Hollywood blockbusters
  - Pioneering techniques in film-making and special effects
  - Representation and diversity in casting and storytelling
  - Reviews and analysis of highly-anticipated new releases
  - The impact of streaming services on movie distribution and consumption
  - The intersection of politics and film, from socially-conscious storytelling to politically-charged controversies
  - Profiles of notable actors, directors, and producers shaping the industry
  - The changing landscape of film festivals and awards season
  - Spotlight on independent and international cinema
  - The legacy of classic films and their enduring cultural impact.

## H.2 Amazon Attributes

### H.2.1 Product Brands

We randomly select 5 categories in Amazon dataset and display the corresponding product brand attributes for each category:

- camera_photo.:
  - Canon
  - Nikon
  - Leica
  - Hasselblad
  - Fujifilm
  - Lumix (Panasonic)
  - GoPro
  - Polaroid
  - Yashica
  - Mamiya
- jewelry_and_watches.:
  - Rolex
  - Cartier
  - Tiffany & Co.
  - Bulgari
  - Omega
  - Patek Philippe
  - Swarovski
  - Gucci
  - Michael Kors
  - Pandora
- magazines.:

- Rolling Stone
- Vogue
- National Geographic
- The New Yorker
- GQ
- Cosmopolitan
- People
- Time
- Sports Illustrated
- Forbes

- health_and_personal_care.:
  - Johnson & Johnson
  - Dove
  - Colgate
  - Head & Shoulders
  - Merck
  - Pfizer
  - Merck
  - Abbott Laboratories
  - GlaxoSmithKline

- toys_games.:
  - Mattel
  - Fisher-Price
  - Hasbro
  - Lego
  - Nerf
  - Barbie
  - Hot Wheels
  - Playmobil
  - MGA Entertainment
  - Paw Patrol

### H.2.2    Product Names

We randomly select 5 categories in Amazon dataset and display the corresponding product name attributes for each category:

- sports_outdoors.:
  - Trekking poles
  - Kayak
  - Stand-up paddleboard
  - Treadmill
  - Bike
  - Yoga mat
  - Weightlifting gloves
  - Athletic training cones
  - Ab wheel
  - Resistance bands
  - Jump rope
  - Rollerskates
  - Boxing gloves

- – Basketball
- – Football
- – Golf clubs
- – Tennis racquet
- camera_photo.:
  - – SnapShooter
  - – FocusReady
  - – ClickCapture
  - – MemoriesMatter
  - – FlashFinder
  - – PicturePerfect
  - – ShotSnap
  - – VisionVibe
  - – PixelPal
  - – FreezeFrame
  - – AngleAttack
  - – SnapStash
  - – FrameFlair
  - – SmileSnaps
  - – ImageImpact
  - – ZoomZest
  - – ClickCraze
  - – PixelPassion
  - – ShootSmart
  - – CaptionChamp.
- grocery_and_gourmet_food.:
  - – Nutella
  - – Doritos
  - – Hellmann's Mayonnaise
  - – Campbell's Soup
  - – Ritz Crackers
  - – Quaker Oats
  - – Ben & Jerry's Ice Cream
  - – Tostitos Salsa
  - – Goldfish Crackers
  - – Red Bull Energy Drink
  - – McCormick Spices
  - – Crystal Light Drink Mix
  - – Funyuns Onion Rings
  - – Skippy Peanut Butter
  - – Heinz Ketchup
  - – Tabasco Hot Sauce
  - – Hershey's Chocolate Syrup
  - – Nescafe Coffee
  - – Kraft Macaroni & Cheese
  - – Gatorade Sports Drink
- baby.:
  - – Baby Swing
  - – Diaper Genie
  - – Milk Warmer

- – Baby Carrier
  - – Car Seat
  - – Baby Monitor
  - – Baby Food Maker
  - – Nursery Glider
  - – Changing Table
  - – Baby Bouncer
  - – Playpen
  - – Teething Rings
  - – Baby Wipes Dispenser
  - – Baby Bath Tub
  - – Bibs
  - – Baby Blankets
  - – Pacifier Clip
  - – Baby Sling
  - – Baby Napper
  - – Moses Basket
- outdoor_living.:
  - – Sunbrella
  - – Weber
  - – Keter
  - – Trex
  - – Solaira
  - – Tropitone
  - – Bimini Solutions
  - – La-Z-Boy Outdoor
  - – Suncast
  - – Beltwood
  - – Quikrete
  - – Cosco Outdoor Living
  - – Anova Furnishings
  - – Duramax
  - – US Polymers
  - – Ostrich Chairs
  - – Carefree of Colorado
  - – Tuff Coat
  - – Fire Sense
  - – Heritage Patios.

## H.3 Reddit Attributes

### H.3.1 Resources

We randomly select 5 categories in Reddit dataset and display the corresponding resource attributes for each category:

- buddy_crossing.:
  - – Meetup.com - a website that allows you to find and join groups of individuals with similar interests in your local area, including hiking, book clubs, and social events.
  - – The Buddy System: Understanding Mental Illness and Addiction - a book that explores the biology of addiction and provides a guide for friends and family members of individuals struggling with these issues.

- Lynda.com - a subscription-based online learning platform that provides courses on a variety of subjects including computer programming, business, web design, and more.
- Codecademy.com - an interactive online platform that teaches coding skills for free or through a subscription.
- Khan Academy - a nonprofit organization that provides free online courses in a wide range of subjects including math, science, and humanities to learners of all ages.
- Duolingo - a language-learning app that is available for free on the App Store and Google Play, offering courses in a variety of languages including Spanish, French, and German.
- MindBody App - a mobile app that helps users find and book local fitness, wellness, and beauty activities such as yoga classes, massages, and haircuts.
- Headspace - a meditation app that offers guided meditation courses to help users reduce stress and improve focus.
- The Knot - a website that provides tools and resources for wedding planning, including a Wedding Website Builder, guest list tracker, and registry management.
- Khan Academy - a nonprofit organization that provides free online courses in a wide range of subjects including math, science, and humanities to learners of all ages.
- Others resource for buddy_crossing.

- the_division.:
  - Division Builds - A subreddit dedicated to sharing and discussing various builds used in The Division.
  - Division Zone - A website with extensive information on game mechanics, gear, and other important gameplay aspects.
  - The Division Discord - A community-run Discord server where players can connect and find groups to play with.
  - The Division Wiki - A comprehensive wiki with guides, tips, and information on everything related to The Division.
  - Skill-Up's YouTube channel - A popular YouTuber who provides detailed analysis and reviews of The Division's updates and patches.
  - MarcoStyle's YouTube channel - Another popular YouTuber who provides in-depth analysis and guides for The Division's gameplay and mechanics.
  - The Division LFG - A website where players can find groups to play with, organize events, and share their experiences.
  - The Division Zone Map - An interactive map that allows players to find important points of interest, loot, and other useful resources.
  - The Division 2 subreddit - A community-run subreddit for the sequel, The Division 2, where players can share their experiences and discuss the game.
  - Others resource for the_division.

- roblox.:
  - Roblox Wiki (https://roblox.fandom.com/wiki/Main_Page)
  - Roblox Developer Forum (https://devforum.roblox.com/)
  - Ultimate Guide to Making Your First Game on Roblox (https://medium.com/@Piranhari/ultimate-guide-to-making-your-first-game-on-roblox-part-1-f1fc63abfbe7)
  - Roblox Blog (https://blog.roblox.com/)
  - Roblox Studio Tutorials (https://www.youtube.com/playlist?list=PLuEQ5BB-Z1SgeZTAAq2w1K3kUfQ-yLEOj)
  - The Roblox Developer Hub (https://developer.roblox.com/en-us/)
  - Top 10 Roblox Games (https://www.techjunkie.com/top-10-best-roblox-games/)
  - Roblox Discord Server (https://discord.gg/roblox)
  - Roblox Support (https://en.help.roblox.com/hc/en-us)
  - Top Roblox Youtubers to Follow (https://www.gamertweak.com/top-roblox-youtubers-to-follow/)

- – Others resource for roblox.
- whats_that_book.:
  - – Goodreads - A social platform for book lovers where users can search for books, create bookshelves, and write reviews.
  - – LibraryThing - A community-driven cataloging website where users can create and share their personal book collections.
  - – AbeBooks - An online marketplace for rare and out-of-print books, as well as other antique or collectible items.
  - – Shelfari - An online book club where users can share book recommendations and read reviews from others.
  - – Project Gutenberg - A digital library of freely available public domain books.
  - – Paperback Swap - A book trading community where users can exchange books with others across the US.
  - – Goodreads Librarians Group - A community of Goodreads users who help with book cataloging, including identifying books from incomplete information.
  - – Book Riot - A website featuring book reviews and book-related news, with an emphasis on diverse and underrepresented voices.
  - – The New York Times Book Review - A renowned weekly publication featuring book reviews, author interviews, and literary criticism.
  - – Others resource for whats_that_book.
- pokemongo_friends.:
  - – Pokemon GO Hub: A comprehensive website dedicated to news, guides, and analysis on Pokemon GO.
  - – The Silph Road Subreddit: A community-run subreddit dedicated to research and analysis of Pokemon GO mechanics.
  - – Poke Assistant: A website that offers a range of tools to help you optimize your Pokemon GO experience, including IV calculators and gym battle simulations.
  - – The Trainer Club: A YouTube channel that provides daily updates, news, and tips for Pokemon GO trainers.
  - – Gotta Catch 'Em All: A Facebook group where you can connect with other Pokemon GO players and coordinate raid battles and other activities.
  - – Reddit's r/PokemonGOFriends Subreddit: A community of players looking for friends to exchange gifts and share invites for raids.
  - – The PokeMap: A website that allows you to find nearby Pokemon on a map in real-time.
  - – Poke Genie: An app that automatically calculates IVs and other stats for your Pokemon, saving you time and headaches.
  - – Pokemon GO Gamepress: A website that offers detailed breakdowns and analysis of Pokemon, movesets, and other game mechanics.
  - – The Go Ranger App: An app that helps you plan your raids and battles, with intuitive mapper tools and filters to help you find the Pokemon you're looking for.
  - – Others resource for pokemongo_friends.

### H.3.2 Experience

We randomly select 5 categories in Reddit dataset and display the corresponding experience attributes for each category:

- build_a_pc.:
  - – DIY PC Builds: Sharing personal experiences and success stories of building custom PCs, discussing component choices, troubleshooting, and performance optimizations.
  - – Budget-Friendly Builds: Discussing experiences with building PCs on a tight budget, sharing cost-saving tips, and recommendations for budget-friendly components.
  - – Cable Management: Sharing personal experiences and tips for effective cable management in PC builds, discussing cable routing techniques and showcasing clean build aesthetics.

- RGB Lighting: Discussing experiences with RGB lighting setups in PC builds, sharing recommendations for RGB components, software customization, and lighting effects.
- Troubleshooting Builds: Sharing experiences and tips for troubleshooting common issues in PC builds, helping fellow builders diagnose and solve hardware or software problems.
- Silent and Quiet PC Builds: Discussing experiences and recommendations for building silent or quiet PCs, focusing on noise reduction techniques and quiet component choices.
- Workstation Builds: Sharing experiences and insights into building PCs for professional workloads, such as video editing, 3D rendering, programming, and graphic design.
- Water-Cooling Adventures: Sharing experiences and insights into custom water-cooling loops, discussing the challenges, benefits, and performance improvements achieved.
- Unique and Custom Builds: Showcasing and discussing unique and custom PC builds, including themed builds, custom cases, or exotic cooling solutions.
- Build Planning and Component Selection: Discussing experiences with planning PC builds, researching and selecting components, considering compatibility, and balancing performance and budget.
- Modding and Case Customization: Sharing experiences with PC case modding and customization, discussing techniques, materials, and showcasing personal projects.
- Compact and Small Form Factor Builds: Discussing experiences with building compact or small form factor PCs, sharing recommendations for mini-ITX cases, cooling solutions, and component choices.
- Home Server and NAS Builds: Sharing experiences and insights into building home servers and network-attached storage (NAS) systems, discussing storage options, software, and data management.
- Multimonitor Setups: Discussing experiences with multimonitor setups, sharing tips for optimizing productivity and gaming experiences across multiple displays.
- PC Gaming Peripherals: Sharing experiences and recommendations for gaming peripherals, such as keyboards, mice, monitors, and headsets, discussing features and personal preferences.

- summon_sign.:
  - Sunbro Covenant: Embracing the Sunbro covenant and assisting fellow players with jolly cooperation, earning sunlight medals and praising the sun together.
  - Fashion Souls: Sharing and showcasing unique and fashionable character builds, armor sets, and weapon combinations for aesthetic enjoyment.
  - Covenant Experiences: Sharing experiences and strategies related to various in-game covenants, such as the Darkwraiths, Blades of the Darkmoon, or Forest Hunters.
  - Community Creations: Showcasing community-created content, such as fan art, videos, or fan fiction, celebrating the creativity and talent within the Summon Sign community.
  - Lore-friendly Builds: Discussing and sharing character builds that are aligned with specific characters or factions within the game's lore, adding immersion and roleplaying elements.
  - Community Appreciation: Expressing gratitude and appreciation for the community, developers, and the overall enjoyment derived from the Dark Souls series and the cooperative multiplayer experiences.

- xbox.:
  - Xbox One exclusive games such as Halo 5, Forza Horizon 4, and Gears of War 4
  - Xbox One media and entertainment apps such as Netflix and Hulu
  - memorable gaming moments or achievements on the Xbox console.
  - Purchase Xbox One online.
  - Xbox Kinect motion sensor accessory
  - Xbox Play Anywhere program
  - Other Experience of Xbox

- pittsburgh.:

- – Visit the Andy Warhol Museum
- – Watch a Steelers football game at Heinz Field
- – Explore the Carnegie Museum of Natural History
- – Ride to the top of Mount Washington on the Duquesne Incline
- – Take a leisurely stroll through Phipps Conservatory and Botanical Gardens
- – Experience the history of the city at the Senator John Heinz History Center
- – Tour the University of Pittsburgh campus
- – Attend a performance at the Benedum Center for the Performing Arts
- – Take a walk along the Three Rivers Heritage Trail
- – Taste pierogies, kielbasa, and other traditional Pittsburgh foods
- – Admire the architecture of the Cathedral of Learning
- – Explore the Pittsburgh Zoo and PPG Aquarium
- – Gaze in awe at the exhibits in the Carnegie Science Center
- – Visit the National Aviary and get up close with tropical birds
- – Check out the local art scene on Penn Avenue in the Garfield neighborhood
- – Attend the Three Rivers Regatta, Pittsburgh's largest annual summer event
- – Take a bike ride on the Great Allegheny Passage trail
- – Ride the roller coasters at Kennywood Amusement Park
- – Discover the nightlife in the South Side neighborhood
- – Go shopping at the Strip District markets for locally-made goods and fresh produce.
- – Others experience for pittsburgh.

- metal_gear_solid.:
  - – Tactical Weapon Customization: Experimenting with various weapons, attachments, and equipment to tailor loadouts to different mission objectives and playstyles.
  - – Character Development: Witnessing the growth and development of iconic characters such as Solid Snake, Big Boss, or Raiden throughout their respective story arcs.
  - – Stealthy Takedowns: Executing silent and non-lethal takedowns, utilizing tranquilizer darts, chokeholds, or sneaking up on enemies from behind.
  - – Tactical Planning: Strategizing mission approaches, analyzing enemy patrols, setting traps, and utilizing distractions to gain the upper hand.
  - – Memorable Characters: Developing connections with unique and memorable characters, such as Otacon, Meryl, Revolver Ocelot, or The Boss.
  - – Stealthy Gadgets: Utilizing gadgets and tools, such as the cardboard box, thermal goggles, or the iconic Solid Eye, to gain advantages during missions.
  - – Emotional Storytelling: Experiencing emotionally impactful moments within the narrative, exploring themes of loss, betrayal, loyalty, and the human cost of warfare.
  - – Espionage Tactics: Participating in undercover missions, gathering intelligence, infiltrating enemy bases, and sabotaging their operations.
  - – Lore and Mythology: Delving into the intricate lore, conspiracies, and historical events within the Metal Gear Solid universe, including topics like The Patriots or the Philosopher's Legacy.
  - – Groundbreaking Game Design: Appreciating the innovative gameplay mechanics, cinematic presentation, and attention to detail that have made the Metal Gear Solid series a beloved and influential franchise in the gaming industry.
  - – Others experience for metal_gear_solid.

## H.4 StackExchange Attributes

### H.4.1 Scenarios

We randomly select 5 categories in StackExchange dataset and display the corresponding scenario attributes for each category:

- multiplayer.:

- Cheating/hacking in online games
- Inappropriate player behavior
- Unbalanced game mechanics
- Difficulty connecting to multiplayer servers
- Matchmaking errors
- Unresponsive or laggy gameplay
- Glitches in gameplay affecting online matches
- Difficulty finding players to match with
- Balancing player skills in matchmaking
- Disconnects and dropped connections mid-game
- Cross-platform compatibility issues
- In-game communication problems
- Difficulty managing and moderating game servers
- Addressing griefing and trolling in multiplayer games
- Managing player accounts and login systems
- Implementing or improving anti-cheat measures
- Community feedback and feature requests
- Addressing game-breaking exploits
- Ensuring fair and accurate reporting of player statistics
- Addressing server crashes and downtime.

- terrain.:
  - Error in generative algorithms for creating terrain
  - Difficulty in implementing procedural terrain generation in a specific game engine
  - Inconsistencies in terrain generation across different devices
  - Issues with realism in terrain generation algorithms
  - Difficulty in implementing terrain physics and collision detection
  - Terrain rendering issues on low-spec hardware
  - Incompatibility between terrain generation and map or level editors
  - Optimization of terrain generation algorithms for speed and memory usage
  - Unwanted artifacts and glitches in terrain mesh generation
  - Compatibility issues between terrain generation algorithms and game engine systems
  - Difficulty in creating realistic terrain textures and materials
  - Inaccuracy of terrain elevation generation in certain geographic regions
  - Difficulty in implementing terrain deformation mechanics
  - Poor performance with large-scale terrain rendering and generation
  - Unwanted noise and roughness in generated terrain meshes
  - Compatibility issues between terrain generation and asset importation pipelines
  - Inaccuracy of terrain heightmap data due to low-quality input data sources
  - Difficulty in handling multi-layer terrain materials and textures
  - Poor performance with dynamic terrain generation and updates
  - Issues with biome and climate-based terrain generation.

- rendering.:
  - Difficulty creating realistic hair and fur in rendering software.
  - Debugging issues with transparent materials in a 3D rendering engine.
  - Crashes or slow performance when rendering large scenes in real time.
  - Trouble with anti-aliasing and other graphics optimization techniques.
  - Struggle with optimizing rendering quality on lower-end hardware.
  - Difficulty incorporating custom shaders into a game engine or rendering pipeline.
  - Figuring out how to use the latest rendering features in a legacy project.
  - Issues with rendering dynamic lighting in real time, without pre-baking.

- Optimizing flicker or aliasing issues caused by fast-moving objects in a scene.
- Solving glitches or crashes caused by malfunctioning GPU drivers.
- Difficulty in rendering complex ocean or water simulations.
- Troubleshooting issues with volumetric rendering in a 3D engine.
- Finding the optimal rendering settings for a particular 3D model or scene.
- Figuring out optimal texturing and lighting in a photorealistic rendering.
- Creating procedural textures and materials in a game engine.
- Debugging flicker issues caused by overlapping or intersecting geometry.
- Difficulty in rendering realistic motion blur in 3D animation.
- Solving imbalanced lighting in a photorealistic rendering of a room or scene.
- Finding the optimal rendering settings for VR or AR applications.
- Debugging issues with inaccurate or glitchy global illumination in a scene.

- procedural-generation.:
  - Improving the efficiency of procedural generation algorithms in Python.
  - Troubleshooting issues with memory usage in large-scale procedural generation projects.
  - Debugging issues with randomized content generation in procedural levels.
  - Implementing procedural generation techniques in C++ for game development.
  - Exploring the potential of machine learning in procedural content generation.
  - Optimizing the generation of complex 3D models using procedural algorithms.
  - Managing complexity and maintaining consistency in procedurally generated game worlds.
  - Addressing issues with procedural generation of text-based content, such as dialogue or item descriptions.
  - Developing tools to aid in the creation and testing of procedural generation algorithms.
  - Balancing the need for randomness with player expectations for fairness and balance.
  - Addressing issues with the procedural generation of music and sound effects.
  - Improving the visual quality of procedurally generated game assets.
  - Exploring ethical concerns around the use of AI in procedural content generation.
  - Developing procedural generation techniques for non-linear narratives or branching storylines.
  - Improving the procedural generation of inhabited environments, such as procedurally generated NPCs.
  - Addressing issues with the procedural generation of terrain features such as rivers and mountains.
  - Implementing procedural generation techniques for user-generated content.
  - Supporting multithreaded execution in procedural generation algorithms.
  - Ensuring procedural generation techniques are compatible with various game engines and frameworks.
  - Improving the scalability of procedural generation algorithms for use in multiplayer games.

- networking.:
  - Difficulty in troubleshooting network connection issues on a Linux system
  - Configuring a wireless access point for a large office space
  - Implementing load balancing across multiple servers in a network
  - Optimizing network throughput to reduce latency in a gaming environment
  - Implementing firewall rules to block unauthorized access to a network
  - Troubleshooting DNS resolution issues on a Windows server
  - Designing and implementing a secure VPN connection
  - Setting up a network file server for shared access among multiple clients
  - Configuring SNMP to monitor network traffic and utilization

- Designing a network topology for a large enterprise with multiple locations
- Troubleshooting issues with Ethernet switches in a data center environment
- Implementing QoS to prioritize network traffic for critical applications
- Configuring NAT and PAT to enable internet access for multiple devices
- Setting up and configuring VLANs to segment a network
- Troubleshooting issues with network printers in an office environment
- Configuring routing protocols in a large network
- Securing wireless access points to prevent unauthorized access
- Troubleshooting issues with VPN connection stability and speed
- Implementing network virtualization with virtual LANs and virtual switches
- Designing and implementing an effective network security strategy to prevent data breaches.

## H.5 ArXiv Attributes

### H.5.1 Subtopics

We randomly select 5 categories in arXiv dataset and display the corresponding product brand attributes for each category:

- hardware_architecture.:
    - Processor design
    - Memory architecture
    - Input/output (I/O) systems
    - Computer networks
    - Digital signal processing
    - Microcontrollers
    - Embedded systems
    - Real-time systems
    - Graphics processing units (GPUs)
    - High-performance computing (HPC)
    - Computer architecture for mobile devices
    - Wearable technology hardware design
    - Secure hardware systems
    - Fault-tolerant hardware design
    - Quantum computing architecture
    - Bio-inspired computing systems
    - Nanoscale computing systems
    - Reconfigurable hardware systems
    - Internet of Things (IoT) devices
    - Green computing and sustainability in hardware design
- genomics.:
    - Genome editing and CRISPR technology
    - Epigenomics and epigenetic modifications
    - Pharmacogenomics and personalized medicine
    - Functional genomics and transcriptomics
    - Metagenomics and microbiome analysis
    - Genomic epidemiology and infectious disease
    - Cancer genomics and tumor heterogeneity
    - Structural genomics and protein structure prediction
    - Plant genomics and crop breeding
    - Animal genomics and livestock improvement

- – Comparative genomics and evolution
  - – Genomic diversity and population genetics
  - – Next-generation sequencing and high-throughput analysis
  - – Single-cell genomics and cellular heterogeneity
  - – Ethical, legal, and social implications of genomics
  - – Cryptography and secure data sharing in genomics
  - – Big data analytics and data mining in genomics
  - – Immune system genomics and immunotherapy
  - – Genomics and aging research
  - – Gene therapy and gene regulation.
- machine_learning.:
  - – Text generation
  - – Natural language understanding for chatbots
  - – Sentiment analysis and opinion mining
  - – Text summarization and keyword extraction
  - – Machine translation
  - – Named entity recognition and entity linking
  - – Dialogue systems and conversational agents
  - – Cross-lingual and Multilingual NLP
  - – Text-to-speech systems
  - – Phonetics and phonology in computational linguistics
  - – Grammatical error detection and correction
  - – Speech recognition and acoustic modeling
  - – Semantic role labeling
  - – Discourse analysis and coherence modeling
  - – Lexical semantics and word sense disambiguation
  - – Computational lexicography and machine-readable dictionaries
  - – Language Modeling
  - – question answering
  - – Language resources and corpora
  - – Computational sociolinguistics and dialectology.
- number_theory.:
  - – Prime numbers
  - – Diophantine equations
  - – Modular arithmetic
  - – Cryptography
  - – Continued Fractions
  - – Pell's Equation
  - – Fermat's Last Theorem
  - – Algebraic Number Theory
  - – Riemann Hypothesis
  - – Arithmetic Geometry
  - – Quadratic Forms
  - – L-Functions
  - – Automorphic Forms
  - – Galois Theory
  - – Ramsey Theory
  - – Distribution of Prime Numbers
  - – Number Theory in Cryptography
  - – Summation Formulas

- – Gaussian Integers
- – The Goldbach Conjecture
- geophysics.:
    - – Seismic imaging
    - – Earthquake prediction
    - – Geothermal energy
    - – Volcanic eruptions
    - – Plate tectonics
    - – Geomagnetism
    - – Paleomagnetism
    - – Geophysical surveying
    - – Geophysical fluid dynamics
    - – Gravity measurements
    - – Rock physics
    - – Crustal deformation
    - – Geomorphology
    - – Mineral exploration
    - – Earth structure modeling
    - – Geodetic techniques
    - – Hydrogeophysics
    - – Earth modeling
    - – Electrical geophysics
    - – Remote sensing geophysics

## H.5.2 Techniques

We randomly select 5 categories in the arXiv dataset and display the corresponding attributes for each category:

- hardware_architecture.:
    - – Binary Decision Diagrams (BDDs) for circuit optimization
    - – Probabilistic CMOS (PCMOS) for energy-efficient computing
    - – Boundary Scan Architecture (BSA) for testing and fault diagnosis
    - – Clock gating for power reduction in synchronous circuits
    - – Memristive networks for neuromorphic computing
    - – Approximate computing for energy-efficient digital processing
    - – Verilog Hardware Description Language (HDL) for circuit design and simulation
    - – Reversible logic circuits for low power consumption
    - – High-level synthesis (HLS) for synthesizing hardware from software descriptions
    - – Phase Change Memory (PCM) for high-density non-volatile storage.
- genomics.:
    - – Genome assembly and annotation using hybrid approaches.
    - – Comparative genomics for analyzing evolutionary relationships between genomes.
    - – Differential gene expression analysis using RNA sequencing data.
    - – Metagenomics for studying the microbial communities in different environments.
    - – Epigenetic analysis for understanding gene regulation.
    - – Network analysis for identifying gene interactions and pathways.
    - – Structural variation analysis for detecting genomic rearrangements.
    - – Functional genomics for studying gene function and pathway regulation.
    - – Genome-wide association studies for identifying genetic variants associated with complex traits.

- High-throughput screening methods for identifying genes involved in specific biological processes.

- computation_and_language.:
  - Word Embeddings
  - Recurrent Neural Networks (RNNs)
  - Attention Mechanism
  - Transformer Model
  - Dependency Parsing
  - Topic Modeling:
  - Machine Translation
  - Corpus Analysis
  - Discourse Analysis
  - Pragmatics

- number_theory:
  - Primality testing using elliptic curves
  - Continued fraction factorization method
  - Algorithm for solving Diophantine equations
  - Quadratic sieve algorithm for integer factorization
  - Pollard rho algorithm for integer factorization
  - Digital sum subtraction method for computing discrete logarithm
  - Fermat's method for factorization of primes
  - Chinese remainder algorithm for solving modular equations
  - Exponential-sum algorithm for computing in algebraic number fields
  - Generalized Ramanujan-Selberg formula for counting integer points on algebraic varieties.

- geophysics.:
  - Seismic attribute interpretation
  - Full waveform inversion
  - Gravity inversion
  - Spherical geometries
  - Ground penetrating radar imaging
  - Time-lapse reservoir monitoring
  - Electrical resistivity tomography
  - Joint inversion of geophysical data
  - Radiometric dating
  - Geomagnetic field modeling

## H.6 AG News Attributes

### H.6.1 Subtopics

We randomly select 5 categories in AG News dataset and display the corresponding subtopic attributes for each category:

- business:
  - Corporate earnings and financial reports
  - Stock market updates and analysis
  - Mergers and acquisitions
  - Business regulations and policies
  - Startups and entrepreneurship
  - Industry trends and forecasts

- – Economic indicators and market trends
- – Business strategies and management practices
- – Corporate governance and ethics
- – Consumer behavior and market research
- – Business leadership and executive profiles
- – Banking and finance industry updates
- – Energy and sustainability in business
- – Retail and e-commerce trends
- – Real estate and property market updates
- – Business disruptions and crisis management
- – Corporate social responsibility and sustainability initiatives
- sci_tech:
  - – Artificial intelligence
  - – Robotics
  - – Quantum computing
  - – Biotechnology
  - – Nanotechnology
  - – Internet of Things
  - – Renewable energy
  - – Virtual reality
  - – Augmented reality
  - – Cybersecurity
  - – Genetic engineering
  - – Big data
  - – Autonomous vehicles
  - – 3D printing
  - – Blockchain technology
  - – Bioinformatics
  - – Machine learning
  - – Biomedical engineering
  - – Clean technology
- sports:
  - – Soccer
  - – Basketball
  - – Baseball
  - – Tennis
  - – Golf
  - – Cricket
  - – Rugby
  - – Athletics
  - – Formula 1
  - – Olympics
  - – Boxing
  - – Swimming
  - – Volleyball
  - – Ice hockey
  - – American football
  - – Cycling
  - – Motorsports
  - – Martial arts

- Horse racing
- Surfing

- world:
    - International politics and diplomacy
    - Global conflicts and war
    - Terrorism and security threats
    - Human rights issues and social justice movements
    - Migration and refugee crises
    - Climate change and environmental policies
    - Global health crises and pandemics
    - Natural disasters and emergencies
    - Cross-border crime and corruption
    - Cultural and social developments worldwide
    - Geopolitical tensions and territorial disputes
    - International aid and development efforts
    - Humanitarian crises and relief efforts
    - Cultural heritage preservation and promotion
    - International collaborations and partnerships

## H.7 SST-2 Attributes

### H.7.1 Subtopics

We randomly select 5 categories in SST-2/IMDB (movie review) dataset and display the corresponding subtopic attributes for each category:

- positive:
    - Compelling Storyline: A strong and engaging narrative that captures the audience's attention from beginning to end.
    - Well-Developed Characters: Memorable and relatable characters that evoke emotions and drive the story forward.
    - Skillful Direction: Effective direction that showcases the filmmaker's vision, ensuring cohesive storytelling and engaging visual elements.
    - Excellent Acting: Convincing performances from the cast that bring the characters to life and immerse the audience in the story.
    - Cinematography: Expertly captured visuals, including the use of framing, lighting, and camera movements, to enhance the storytelling and create a visually appealing experience.
    - Engaging Dialogue: Well-written dialogue that is natural, meaningful, and contributes to character development and plot progression.
    - Sound Design and Music: Thoughtful and immersive sound design, including sound effects and a well-curated soundtrack or original score, that enhances the overall cinematic experience.
    - Production Design: Attention to detail in creating visually appealing and authentic sets, costumes, and overall aesthetics that contribute to the film's atmosphere and world-building.
    - Editing: Skillful editing that maintains a good pace, effectively transitions between scenes, and enhances the overall flow and impact of the story.
    - Emotional Impact: A movie that evokes emotions, whether it be through humor, drama, suspense, or other means, leaving a lasting impression on the audience.
- negative:
    - Weak Plot: A poorly developed or uninteresting storyline that fails to engage the audience.

- **Lackluster Performances**: Unconvincing or uninspired performances by the actors that fail to bring the characters to life.
- **Poor Production Quality**: Subpar production values, including low-quality visuals, amateurish cinematography, and weak special effects.
- **Incoherent Storytelling**: Confusing or disjointed narrative structure that makes it difficult to follow or understand the plot.
- **Unmemorable Characters**: Underdeveloped or forgettable characters that fail to resonate with the audience.
- **Weak Soundtrack**: A forgettable or poorly composed soundtrack that fails to enhance the mood or add depth to the movie.
- **Poor Dialogue**: Uninteresting or poorly written dialogues that fail to engage or resonate with the audience.
- **Disjointed Atmosphere**: A lack of coherence or consistency in creating an immersive and believable world for the viewers.
- **Unresolved Plotlines**: Loose ends or unresolved plotlines that leave the audience feeling unsatisfied or confused.
- **Lack of Entertainment Value**: A movie that fails to deliver an enjoyable or engaging experience for the audience, leaving them feeling bored or uninterested.

### H.7.2 Descriptive Details

We use movie genres as the characteristics of movies, and the attributes are listed as follows:

- Action
- Drama
- Comedy
- Thriller
- Romance
- Horror
- Adventure
- Science Fiction
- Fantasy
- Animation

## H.8 Yelp Attributes

### H.8.1 Subtopics

We randomly select 5 categories in Yelp review dataset and display the corresponding subtopic attributes for each category:

- positive:
  - **Quality of Food**: The taste, flavor, and presentation of the dishes.
  - **Fresh Ingredients**: The use of fresh and high-quality ingredients in the preparation of the food.
  - **Menu Variety**: A diverse range of options catering to different dietary preferences and restrictions.
  - **Presentation**: The visually appealing presentation of the dishes.
  - **Service**: Attentive, friendly, and prompt service from the restaurant staff.
  - **Value for Money**: Offering good quality and portion sizes at reasonable prices.
  - **Cleanliness**: A clean and well-maintained dining area, including tables, utensils, and restrooms.
  - **Special Dietary Accommodations**: Catering to specific dietary needs such as vegetarian, vegan, gluten-free, etc.

- – Unique and Creative Dishes: Offering innovative and creative dishes that stand out.
- – Efficient Operations: Smooth and well-coordinated operations to minimize waiting times and delays.
- negative:
  - – Poor Service: Slow or inattentive service from the restaurant staff. Unfriendly Staff: Rude or unhelpful behavior from the restaurant staff.
  - – Long Waiting Times: Excessive waiting times for a table or food.
  - – Incorrect Orders: Receiving incorrect or poorly prepared food orders.
  - – Unappetizing Presentation: Dishes that are poorly presented or lack visual appeal.
  - – Unpleasant Ambience: Uncomfortable or uninviting atmosphere in the restaurant.
  - – Dirty or Unhygienic Conditions: Lack of cleanliness in the dining area, restrooms, or utensils.
  - – Limited Menu Options: A limited selection of dishes or lack of variety.
  - – Poor Food Quality: Dishes that are poorly cooked, tasteless, or of low quality.
  - – Overpriced: Excessive prices for the quality and portion sizes of the food.

### H.8.2 Descriptive Details

We use cuisine types as the characteristics of restaurants, and the attributes are listed as follows:

- Turkish
- Spanish
- Greek
- Italian
- French
- American
- Mexican
- Canadian
- Cajun
- Tex-Mex
- Brazilian
- Peruvian
- Argentinean
- Colombian
- Venezuelan
- Ethiopian
- Moroccan
- South African
- Nigerian
- Egyptian
- Chinese
- Japanese
- Indian
- Thai
- Korean
- Australian
- New Zealand
- Polynesian
- Hawaiian
- Singaporean

# I Examples for Filtered Attribute Values

Here we give some examples of the filtered attributes.

For the Amazon product review dataset, some filtered attributes are listed as follows.

- beauty:
    - Hair Dryer (close to health and personal care)
    - Hair Straightener (close to health and personal care)
- electronics:
    - Car dashcam (close to automotive)
    - Wireless earbuds (close to cell_phones_service)
- office_products:
    - Mouse pad (close to electronics)

For NYT dataset, some filtered attributes are listed as follows:

- american_football:
    - The economic impact of football on local communities and businesses.
    - The role of nutrition and hydration in optimal football performance
    - The history of football's integration with America's armed forces
- stocks_and_bonds:
    - The role of artificial intelligence in stock market prediction
    - The effects of climate change on bond performance
- movies:
    - The future of cinema in a post-COVID world
    - The role of international markets in Hollywood's global success

For Reddit dataset, some filtered attributes are listed as follows:

- amc_stock:
    - Working in the stock market industry.
    - Impact of COVID-19 on amc_stock.
    - Having a background in finance or business.
- what_car_should_i_buy:
    - Towing capacity and trailer options
    - Financing and leasing options
    - Road trip and travel experiences
- pittsburgh:
    - Attending a game at PNC Park
    - Watch a Steelers football game at Heinz Field

For StackExchange dataset, some filtered attributes are listed as follows:

- algorithm:
    - Debugging an optimization algorithm for a complex logistics problem.
    - Improving the robustness of an encryption algorithm against attacks.
    - inding a way to better handle overflow errors in a cryptographic algorithm.
- iPhone:
    - Unable to use certain features in certain regions/countries
    - Trouble with GPS not working properly
- opengl:
    - Difficulty setting up a custom camera in OpenGL
    - Inconsistent rendering of 3D models in OpenGL
    - Difficulty integrating OpenGL with a cross-platform game engine

# J Examples for Merged Attribute on the arXiv dataset

Here we give some examples of the filtered attributes.

- optimization techniques for computer vision:
  - computer vision and pattern recognition
  - machine learning

- black holes:
  - classical physics
  - popular physics

- evolving networks for financial trading:
  - neural and evolutionary computing
  - computational finance

- multi-lingual information retrieval:
  - information retrieval
  - computation and language

- iot operator systems and their security challenges:
  - operating systems
  - cryptography and security

- complexity of social networks:
  - computational complexity
  - social and information networks

# K The Prompt Format for MetaPrompt

In this section, we provide more details about the implementation of MetaPrompt [45]. It first leverages LLMs to provide additional task-specific information, and then augments the class-dependent prompt with the task-specific knowledge.

**Prompts to ChatGPT**: `[Original Prompt]`. What does this task ask us to do?

**Response from ChatGPT for NYT dataset**: The task of generating an example of a NYT news asks the model to create high-quality, informative, and engaging news articles that mimic the writing style and structure commonly found in The New York Times (NYT).

**Response from ChatGPT for Amazon dataset**: The task of generating an example of a product review asks the model to create a detailed and informative review for a particular product, highlighting its features, performance, pros and cons, and overall user experience. This review should be helpful for potential buyers, providing them with valuable insights to make an informed purchasing decision.

**Response from ChatGPT for Reddit dataset**: The task of generating an example of a post similar to Reddit asks the model to craft a well-written and engaging post that adheres to the guidelines and community standards of the specific subreddit it's intended for. The post should spark discussions, share valuable information, or showcase interesting content that resonates with the subreddit's audience.

**Response from ChatGPT for StackExchange dataset**: The task of generating an example of a post similar to StackExchange asks the model to compose a clear and concise question or answer related to a specific topic within the given StackExchange community. The post should be well-researched and supported by relevant sources or personal expertise, aiming to provide valuable insights or solutions to other users' queries.

# L  Dataset Information

## L.1  Dataset Documentations

The dataset is provided in *json* format; there are three json files corresponding to the original `train`, `validation` and `test` split. We also include two additional file, `attrprompt` and `simprompt`, which is generated by AttrPrompt and SimPrompt as the synthetic training data.

Each data point contains the following fields:

- `label`: the label for the example. For multi-class classification, the `label` field is an integer, while for multi-label classification, the `label` field is a `list[int]` containing one or multiple integers as each example may refer to multiple classes;
- `text`: a content of each example.

## L.2  Intended Uses

AttrPrompt and SimPrompt are intended for researchers in machine learning, natural language processing, and related fields to innovate novel methods for training data generation problems.

## L.3  Hosting and Maintenance Plan

The codebase is hosted and version-tracked via GitHub. It will be available under the link `https://github.com/yueyu1030/attrprompt`. The download link of all the datasets can be found in the Github repository.

Note that it is a community-driven and open-source initiative. We are committed and have the resources to maintain and actively develop it for at minimum the next five years. We plan to grow the GitHub repo by including new tasks and datasets and warmly welcome external contributors.

## L.4  Licensing

We license our work using Apache 2.0[8]. All of the `train/validation/test` data are publicly released by previous work [1, 32, 17].

## L.5  Author Statement

We the authors will bear all responsibility in case of violation of rights.

---

[8]https://www.apache.org/licenses/LICENSE-2.0

