# OpenReview forum: "Large Language Model as Attributed Training Data Generator: A Tale of Diversity and Bias"
_NeurIPS.cc/2023/Track/Datasets_and_Benchmarks — NeurIPS 2023 Datasets and Benchmarks Poster_

### Official Review · Reviewer_upz4 · 2023-07-20

**Rating:** 7
**Confidence:** 3
**Correctness:** Yes
**Clarity:** Yes

**Strengths:**

It applies the paradigm of LLM-as-training-data-generator to multiple challenging datasets with higher cardinality compared to datasets used in prior work; it also takes the first step to apply such paradigm on multi-label classification task
This study delves into two important aspects of data generated by LLM, ie, diversity and bias, and reveal that existing class-conditional prompt could lead to severe bias in the generated data, which can be alleviated by attributed prompt
Empirical results show that attributed prompts not only have better model performance, but also better sample/budget efficiency




**Additional Feedback:**

It could be better if more comprehensive experiments can be conducted on more LLMs.

**Documentation:**

The absence of versioning makes it challenging to monitor any potential future updates to the dataset, limiting the possibility of further community contributions.

**Limitations:**

Yes

**Opportunities For Improvement:**

I don’t see any major weakness of this study, but one interesting future work would be to optimize the attribute configuration for best downstream performance. The code accompanying the paper would benefit from a better integration into standard APIs.




**Relation To Prior Work:**

Yes, to the best of my knowledge, related work is discussed adequately.

**Summary And Contributions:**

This paper studies a recent data-centric AI paradigm of LLM-as-training-data-generator. In particular, it explores using diversely attributed prompts for training data generation. It shows that such attributed prompts could help reduce the bias in the generated dataset and enhance its diversity. On multiple datasets, empirical results demonstrate the superiority of using attribute prompts over simple class-conditional prompts.

---

> ### Author Response · Authors · 2023-08-22
>
> We would like to thank the reviewer for the useful feedback. Here we provide a response to your concerns:
>
> > One interesting future work would be to optimize the attribute configuration for best downstream performance. The code accompanying the paper would benefit from a better integration into standard APIs.
>
> Thanks for this helpful comments. Overall we believe it is often hard to optimize the attribute configurations under the challenging zero-shot learning setting due to the lack of task-specific knowledge. That being said, if there is a small set of labeled data available for the target task, we could possibly use them to determine the attributes as well as the distribution of different values for each attribute and conduct a weighted sampling to mimic the target distribution. We further discuss this issue in *Appendix A in the revised manuscript*.
> Thanks for your suggestions, and we will strive to optimize the code structure to better integrate them to standard APIs.
>
>
> > The absence of versioning makes it challenging to monitor any potential future updates to the dataset, limiting the possibility of further community contributions.
>
> Thanks for pointing this issue out! We have added the versioning of the generated data based on your suggestion.
>
> > It could be better if more comprehensive experiments can be conducted on more LLMs.
>
> Thanks for your insightful review! Please see our general response for details.

---

### Official Review · Reviewer_mJxq · 2023-07-22
**A Good Approach to Data Generation Using LLM and Dataset Meta Information**

**Rating:** 6
**Confidence:** 4
**Correctness:** The paper conducts a lot of experimen…

**Strengths:**


1. [Novel meta-info leveraging] The paper explores an innovative approach - using diversely attributed prompts instead of standard class-conditional prompts for generating training data, which is a relatively less explored area in the domain of large language models.

2. [Performance improvement] It demonstrates that attributed prompts outperform simple class-conditional prompts in terms of the resulting model's performance.

3. [Cost efficiency] The study shows that attributed prompts achieve the same performance as class-conditional prompts while reducing the querying cost by 95%.

4. [Comprehensive experiments and analysis] It offers a comprehensive empirical study on attributes such as bias, diversity, and efficiency in the generated data, providing a more in-depth understanding of potential issues and solutions in training data generation.


**Additional Feedback:**

Plz see details above

**Clarity:**

This paper is well-written and easy to follow. The authors clearly presents their contribution and the significance of their work, providing good readability.

**Documentation:**

This work provides sufficient information in terms of data collection, organization, availability and maintenance, together with a URL that is publicly accessible.

**Opportunities For Improvement:**


1. [Specificity & limited model scales] The findings are based on *classification* datasets with high cardinality and from diverse domains. Besides, the adopted models are a bit out-of-date and small (14M ~ 435M BERTs). From Figure 5, it seems that the performance improvements have a tendency to stagnate with larger model sizes. These results might not be generalizable to other task types and larger models.

2. [Bias Analysis] Although the paper acknowledges the presence of biases in synthetic datasets generated by simple prompts, there is less focus on how to mitigate such biases.

3. [Experiment Replicability] The performance brought by the method might depend heavily on the proper selection of attributed prompts, which might require expert knowledge and make the method less accessible to non-experts.

4. [Oversimplification] The proposed method is heuristic and relatively simple. It generally assumes that diversely attributed prompts yield more diverse and less biased data, which ignores potential complexities in how attributes interact or biases inherent in the attributes themselves.


**Relation To Prior Work:**

This work provides clear explanation to how it differs from the existing contributions in their related work section

**Summary And Contributions:**

This paper investigates the use of large language models (LLMs) as training data generators and the effect of using different prompts when generating this data. The research notes that while LLMs have been used for various natural language processing tasks, reliance on simple class-conditional prompts may limit the data's diversity and introduce biases.

To overcome this, the authors propose using diversely attributed prompts, which can specify attributes like length and style. Applied to datasets with high cardinality and from diverse domains, the results illustrate that attributed prompts outperform simple class-conditional prompts in terms of the resulting model's performance.
Besides, extensive empirical study are conducted and some insights are provided, e.g., attribute diversity can enhance model performance.

In a nutshell, its contributions mainly include: (1) a novel approach for training data generation using diversely attributed prompts that improves both model performance and data efficiency; and (2)a comprehensive study on the impact of prompt types on bias and diversity in generated data.

---

> ### Author Response · Authors · 2023-08-22
> **Response 1/2**
>
> Thanks for your thoughtful feedback! We discuss your raised points as follows:
>
> > [Specificity & limited model scales] The findings are based on classification datasets with high cardinality and from diverse domains. Besides, the adopted models are a bit out-of-date and small (14M ~ 435M BERTs). From Figure 5, it seems that the performance improvements have a tendency to stagnate with larger model sizes. These results might not be generalizable to other task types and larger models.
>
> Thanks for your thoughtful comments. We would like to mention that these classification tasks with high cardinality are still challenging: as shown in our experiments, under the fine-grained classification setting, there is still 5%-30% performance gap between the performance of real training data and synthetically generated data (with the same size). We agree that generalizing Attrprompt would be an important avenue of future work, and we add more discussion on *Appendix A in the revised manuscript* to better reflect the scope of our paper.
>
> For the **model size**, the rationale to employ models in the 14M ~ 435M BERTs range is in twofold. Firstly, these models, while seemingly "out-of-date" to some, are still widely utilized in numerous real-world applications due to their balance between computational cost and performance. Secondly, by evaluating these "smaller" models, we aimed to understand the gains in performance relative to the model's size. That being said, we have attempted to run the experiments using T5-Large, a pretrained text-to-text transformer model with 770M parameters, the results are shown in the following table:
>
>
> |    Dataset        | NYT   | Amazon | Reddit | StackExchange |
> |------------|-------|--------|--------|---------------|
> | SimPrompt  | 85.68 | 63.80   | 60.20   | 40.86         |
> | AttrPrompt | 87.46 | 72.73  | 72.11  | 45.23         |
>
> Overall, we observe that the gain of AttrPrompt still exists for all datasets, and there are performance improvements over the second large model (DeBERTa-v3-large) especially on Amazon (3.9%) and Reddit (2.3%) dataset.
>
> ## References:
> [1] Meng et al. Generating training data with language models: Towards zero-shot language understanding. Neural Information Processing Systems 35 (2022): 462-477.
>
> [2] Ye et al. ZeroGen: Efficient Zero-shot Learning via Dataset Generation. Conference on Empirical Methods in Natural Language Processing. 2022.
>
> [3] Gao et al. Self-Guided Noise-Free Data Generation for Efficient Zero-Shot Learning. International Conference on Learning Representations. 2023.
>
> > [Bias Analysis] Although the paper acknowledges the presence of biases in synthetic datasets generated by simple prompts, there is less focus on how to mitigate such biases.
>
> Thank you for pointing out the need for clarity on bias mitigation. Our method is primarily devised to counteract the biases found in training data stemming from simple class-dependent prompts, as it aims to equilibrate the distribution across various attribute values. Nevertheless, we would like to note that our approach is optimized for a zero-shot setting where the ground-truth distribution remains unobserved. When we have access to task-specific corpora, a feasible strategy could involve leveraging LLM to identify attributes and their respective distributions within the target domain to further alleviate the bias.

---

> ### Author Response · Authors · 2023-08-22
> **Responses 2/2**
>
> > [Experiment Replicability] The performance brought by the method might depend heavily on the proper selection of attributed prompts, which might require expert knowledge and make the method less accessible to non-experts.
>
> Thank you for bringing up the important point of experiment replicability. Indeed, the effectiveness of our method can be influenced by the selection of attributes, but in our study, we note that the selection of the attributes is not so hard — we conduct an addition experiment over four human raters who do not have the specific expertise on the evaluation tasks. The overall time for human raters to perform the selection is very few — often within 1 minutes, and the Fleiss’ Kappa over the selection of raters are 0.81 (NYT), 0.683 (Amazon), 0.798 (Reddit) and 0.625 (StackExchange) — all of the scores fall into the category of “Substantial Agreement” or “Near Perfect Agreement”, and indicates that the selection does not require *specific expertise* and can be *consistent* across individuals.
>
> Furthermore, we conducted supplementary experiments on two datasets (NYT and Amazon), where we generated training data using all attributes *without filtering*. The performance is shown in the following table:
>
>
> |                      | NYT   | Amazon |
> |----------------------|-------|--------|
> | GPT-3.5 Zero-shot      | 69.84 | 54.56  |
> | SimPrompt            | 76.34 | 56.96  |
> | AttrPrompt           | 82.26 | 65.87  |
> | AttrPrompt w/o selection | 81.08 | 63.76  |
>
> Although there is a little performance drop, the performance is still *significantly better than the SimPrompt baseline*, indicating that the performance of AttrPrompt is relatively robust to the irrelevant attributes. We have added more discussion in **Appendix E.2. in the revised manuscript** on this issue.
>
> > [Oversimplification] The proposed method is heuristic and relatively simple. It generally assumes that diversely attributed prompts yield more diverse and less biased data, which ignores potential complexities in how attributes interact or biases inherent in the attributes themselves.
>
> Thanks for pointing out this potential issue. We consider being simple and effective as an advantage. We believe simple solutions are more robust, easier to generalize, and hopefully can inspire more future research explorations on designing better LLM prompting approaches for training data generation.
> For the interaction and biases in the bias, in fact, we *do not assume* that diversely attributed prompts yield more diverse and less biased data but *empirically show* that attributed prompts could yield more diverse and less biased data in section 4. We acknowledge that our approach cannot eliminate bias entirely or account for more complex interactions between attributes, but by incorporating different attribute combinations, our approach can actually mitigate the bias on attribute distributions (as shown in section 4.3). One of the important avenues for future works could be leveraging LLMs or human-AI collaboration frameworks to further identify biases/higher-order interactions among attributes. We have added more discussion on the “Limitation” and “Future Work” sections in **Appendix A of the revised manuscript** to discuss this issue.
>
> Thanks again for your review! Please let us know if you have any further questions.

---

### Official Review · Reviewer_KfWS · 2023-07-23
**Initial review**

**Rating:** 7
**Confidence:** 4
**Correctness:** Enhancing prompts by adding attribute…
**Clarity:** The presentation of this paper is cle…

**Strengths:**

- Using language models to generate data for further use is an interesting and useful problem. How to design more powerful prompts that unlock the strong capability of LLMs for more useful applications is a decisive direction.

-  The distribution of cosine similarity of text pairs is very informative and intuitive, clearly showing how having attributes in the prompts help to improve the generated text.

- The experiments are

**Additional Feedback:**

Please check the previous sections.

**Documentation:**

The documentation is well-organized and clear.

**Ethics:**

I do not notice any ethics concerns.

**Limitations:**

The authors discuss their limitations and potential negative societal impact  in appendix.

**Opportunities For Improvement:**

- The major improvement of the prompts in this paper is to incorporate attributes in the prompts. Althought the authors discuss what attributes can be used in the four datasets in this paper, the attributes in general are very specific to datasets. Is there any principle or insights of what attributes should be selected generally?

- In the paper,  4-5 attributes are used for each dataset. They work pretty well for the datasets used in this paper. But for a more general purpose, how many attributes should be added in general?

- The language models used in this paper are Bert series models. The authors could consider apply the attributed  prompts to most recent sota LLMs such as LLaMA, or if possible, non-open models like gpt.

- Since this is a dataset and benchmark track, this paper mainly proposes an improvement for data generation using LLMs. I have little concerns about whether this is in the scope of this track. I do not think this is a big issue but flag here and hope the authors could clarify this.

**Relation To Prior Work:**

To the best of my knowledge, the prior work is well discussed.

**Summary And Contributions:**

Using language models to generate data has many applications. This paper claims that only using the class labels in the prompts may lead to biased generated data. To this end, the authors propose to enhance the promotes by adding specific attributes such that the LMs can generate more specific data for any kind of attribute. The proposed new attributed-enhanced prompts are evaluated in several datasets, and show clear improvements comparing to only using  class labels.

---

> ### Author Response · Authors · 2023-08-22
> **Responses 1/2**
>
> We would like to thank the Reviewer for the useful feedbacks. We have answered your questions in the response below.
>
> > The major improvement of the prompts in this paper is to incorporate attributes in the prompts. Although the authors discuss what attributes can be used in the four datasets in this paper, the attributes in general are very specific to datasets. Is there any principle or insights of what attributes should be selected generally?
> > In the paper, 4-5 attributes are used for each dataset. They work pretty well for the datasets used in this paper. But for a more general purpose, how many attributes should be added in general?
>
> Thanks for these insightful questions.  We answer your questions as follows:
>
> **For the selection of attributes**: To select appropriate attributes, a general rule-of-thumb for any datasets could be *promoting diversity* (e.g. text style), as well as *reduce bias*, as it should identify important attributes (such as demographic information) to prevent the model from generating skewed datasets. On the other hand, We also acknowledge that for each target dataset, some *dataset-specific considerations* are indeed needed (e.g. ‘subtopics’ for News and ‘product names’ for Amazon Product Review) to tailor the specific data distribution  so that the model trained on the synthetic data generated with the attribute can generalize well.
>
> **For the number of attributes**: The number of attributes shouldn't be too large or too small. Too few attributes can limit dataset diversity, while an excessive number can lead to an overwhelming array of attribute combinations. If the combinations surpass the size budget, the ability to enumerate all combinations diminishes, potentially impacting generalization.
>
> **Attribute Selection for Four Datasets in this study**: It is also worth noting that for the four datasets studied in the paper, there is high agreement among human raters for attribute selection — we conduct an addition experiment over four human raters, and the Fleiss’ Kappa over the selection of raters are 0.81 (NYT), 0.683 (Amazon), 0.798 (Reddit) and 0.625 (StackExchange) — all of the scores fall into the category of “Substantial Agreement” or “Near Perfect Agreement”, and indicates that the selection can be consistent across individuals.
>
> **Furthur Discussion on Attribute Selection**: In this work, we aim to demonstrate the usefulness of attributes and why they are useful through the lens of diversity and bias, and we think this opens the door for future work to further explore the methodology for attribute discovery and selection for optimal performance. We concede that it could be difficult to give a formal, mathematical principle to automatically determine the optimal number/property of attributes *without any labeled data and task-specific information*. However, if a small labeled dataset is available, leveraging Language Models can aid in automatically detecting diverse attributes for each class, which can further reduce the human efforts for selecting key attributes. We have follow your suggestion to add more discussion on this issue in the ‘limitation’ and 'future work' section in the **Appendix A of the revised manuscript**.

---

> ### Author Response · Authors · 2023-08-22
> **Responses 2/2**
>
> > The language models used in this paper are Bert series models. The authors could consider apply the attributed prompts to most recent sota LLMs such as LLaMA, or if possible, non-open models like gpt.
>
> **Clarification on Language Models and Experimentation**: We would like to clarify that in this work, there are two types of language models: (1) **the generative LLMs** (ChatGPT used in main experiments) which generate training data with the input prompts; (2) **the language model classifiesr** (BERT series models used in experiments), which are finetuned on the generated training data to perform classification on downstream tasks. While these classification models seem to be "out-of-date", they are still widely utilized in numerous real-world applications due to their balance between inference cost and performance.
>
> **The rationale for not using LLM in-context learning for classification**: We have also attempted to leverage LLM in-context learning paradigm as the downstream classifier, but find it hard for the classification task with higher cardinality — a common few-shot scenario (with k=5) will result in more than 100 in-context demonstrations for tasks studied in this work. As a result, it would be hard to fit them all into the LLMs, and we believe that in-context learning with high-cardinality classification and synthetic data merits a publication on its own.
>
> **Additional Experiments on Classifiers**: That being said, we have **added more experiments** to demonstrate the effectiveness of Attrprompt on classifier models other than BERT-series models: we have attempted to run the experiments using **T5-Large**, a pretrained text-to-text transformer model with 770M parameters, and the results are shown in the following table:
>
>
> |    Dataset        | NYT   | Amazon | Reddit | StackExchange |
> |------------|-------|--------|--------|---------------|
> | SimPrompt  | 85.68 | 63.80   | 60.20   | 40.86         |
> | AttrPrompt | 87.46 | 72.73  | 72.11  | 45.23         |
>
> Overall, we observe that the gain of AttrPrompt still exists for all datasets.
>
> **Additional Experiments on LLM Generators**: Besides, we have also conducted additional experiments using the LLaMA-2 and other GPT models as generators. Please see our general response.
>
>
> > Since this is a dataset and benchmark track, this paper mainly proposes an improvement for data generation using LLMs. I have little concerns about whether this is in the scope of this track.
>
> We appreciate the reviewer's concern about the scope of our paper. Please see the general response for the explanations.
>
> Thanks again for your feedbacks! Please let us know if you have any further questions.

---

> > ### Comment · Reviewer_KfWS · 2023-08-22
> > **Thanks for the rebuttal**
> >
> > I appreciate the authors for their efforts in addressing my questions. My questions and concerns are mostly addressed. Please include the new clarification and results in the next version. I increase my score to 7.

---

> > > ### Author Response · Authors · 2023-08-22
> > >
> > > Thanks for your further feedback as well as raising the score! We are glad that our rebuttal addressed most of your concerns. We will surely include these clarifications and new experimental results in the next version of the manuscript.

---

### Official Review · Reviewer_rTXq · 2023-07-26
**Large Language Model as Attributed Training Data Generator: A Tale of Diversity and Bias**

**Rating:** 7
**Confidence:** 4
**Correctness:** Yes, some model beyond ChatGPT will m…
**Clarity:** Yes

**Strengths:**

- interesting and timely problem statement:  diversity of the generated data and inherit systematic biases of LLM.
- investigation focusing on datasets with high cardinality and diverse domains.
- interesting results over the baselines.
- interesting analysis and observations on bias, diversity, and efficiency.
- well written paper.

**Additional Feedback:**

NA

**Documentation:**

Yes

**Ethics:**

No.
Since authors use ChatGPT for data generation, I hope the released license of the dataset is in compliance with OpenAI guidelines for their API usage. Authors may want to clarify this further.

**Limitations:**

Yes, See "Opportunities for Improvement" for making this part better.

**Opportunities For Improvement:**

- "we ground the LLM to ChatGPT": While ChatGPT is a great model, it is important to show efficacy of this method on other models. Irrespective of the results (positive or negative), that would complete the analysis. I am not sure if the current evaluation with ChatGPT is sufficient analysis for the method author propose in the paper.
- [Not a limitation, but scope for improvement] Based on the interesting results, I see scope in this work to also do data quality analysis independent of downstream task performance. While vocabulary is a good measurement, there are other interesting parameters such as in https://arxiv.org/abs/2009.10795 and https://arxiv.org/abs/2005.00816) authors might want to discuss.
- [Question]:
How sensitive is the data generation process to the instructions? Does reframing instructions (e.g. https://aclanthology.org/2022.findings-acl.50.pdf, https://arxiv.org/abs/2102.07350) help improve downstream model performance? This is important to reproduce the same method on other data generation tasks and study its robustness.
- [Question]: L112: "manually select the attribute dimensions of the highest quality that best suit the dataset.": how do you do this? I hope this does not produce some sort of test data feature leakage? and also not introduce additional human bias in this process?


**Relation To Prior Work:**

More instruction learning/prompting references need to be discussed, since the data generation approach is primarily based on instructing LLM to generate data.

For data generation, the following paper is relevant and needs to be discussed
https://arxiv.org/abs/2201.05955

**Summary And Contributions:**

This paper studies an important aspect of data generation with LLMs: diversity of the generated data and inherit systematic biases of LLM.

- They investigate training data generation with diversely attributed prompts (e.g. specifying attributes like length and style), which have the potential to yield diverse and attributed generated data.
- They demonstrate that attributed prompts outperform simple class-conditional prompts in terms of the resulting model’s
performance.
- They conduct analysis on vital aspects like bias, diversity, and efficiency.

---

> ### Author Response · Authors · 2023-08-22
> **Responses 1/2**
>
> Thanks for your thoughtful feedback! We discuss your raised points as follows:
>
> > "We ground the LLM to ChatGPT": While ChatGPT is a great model, it is important to show efficacy of this method on other models. Irrespective of the results (positive or negative), that would complete the analysis. I am not sure if the current evaluation with ChatGPT is sufficient analysis for the method author propose in the paper.
>
> Thanks for this suggestion. Please see our general response for experiments on other LLMs.
>
> > [Not a limitation, but scope for improvement] Based on the interesting results, I see scope in this work to also do data quality analysis independent of downstream task performance. While vocabulary is a good measurement, there are other interesting parameters such as in https://arxiv.org/abs/2009.10795 and https://arxiv.org/abs/2005.00816) authors might want to discuss.
>
> Thanks for mentioning these related papers! Following your suggestion, We have added two additional metrics, namely *average pairwise cosine similarity* (APS) and *Inter-sample N-gram Frequency (INGF)*  in paper [1] to further quantify the diversity.
> For APS, the *lower* stands for better diversity. For INGF, the *higher* stands for better diversity.  The result is shown as follows:
>
>
> |            |      |    **NYT**   |      |                 |  |    **Amazon**   |      |                 |
> |------------|--------------|------|------|-----------------|-------------|------|------|-----------------|
> |            | Inter-Class APS  | Intra-Class APS | All Sample APS | N-gram Frequency | Inter-Class APS | Intra-Class APS | All Sample APS  | N-gram Frequency |
> | Simprompt  | 0.101        | 0.568| 0.135| 5277.2          | 0.207       | 0.620| 0.241| 2266.5         |
> | Attrprompt | 0.159        | 0.474| 0.182| 6688.6          | 0.225       | 0.483| 0.246| 2605.5         |
> | Gold       | 0.098        | 0.358| 0.122| 7618.1          | 0.101       | 0.251| 0.114| 4992.1       |
>
> |            |    |    **Reddit**  |      |                 |  |   **StackExchange**   |      |                 |
> |------------|--------------|------|------|-----------------|-------------------|------|------|-----------------|
> |            | Inter-Class APS | Intra-Class APS | All  | N-gram Frequency | Inter-Class APS    | Intra-Class APS | All Sample  APS | N-gram Frequency |
> | Simprompt  | 0.173        | 0.818| 0.201| 2697.8          | 0.282             | 0.804| 0.302| 2259.8          |
> | Attrprompt | 0.106        | 0.474| 0.122| 3994.5          | 0.105             | 0.375| 0.114| 2464.3          |
> | Gold       | 0.044        | 0.261| 0.054| 9079.6          | 0.056             | 0.196| 0.063| 5492.4          |
>
> Overall, we observe that under these two new metrics, AttrPrompt achieves better diversity than Simprompt, which further justifies our claims. We have added additional experimental results and discussions in table 6, section 4.1 in the revised manuscript.
>
> > [Question]: How sensitive is the data generation process to the instructions? Does reframing instructions (e.g. https://aclanthology.org/2022.findings-acl.50.pdf, https://arxiv.org/abs/2102.07350) help improve downstream model performance? This is important to reproduce the same method on other data generation tasks and study its robustness.
>
> Thanks for this insightful review. Following your suggestion, we have (1) added more discussion on these prompt reframing approaches in **section 2 of the revised manuscript**, (2) added one additional baseline named *MetaPrompt* [2], which uses a large language model itself to optimize the input prompt. Please see our *general response* for details.
>
> ## References
> [1] Mishra et al. "Dqi: Measuring data quality in nlp." arXiv preprint arXiv:2005.00816 (2020).
>
> [2] Reynolds and McDonell. "Prompt programming for large language models: Beyond the few-shot paradigm." arXiv preprint arXiv:2102.07350 (2021).

---

> ### Author Response · Authors · 2023-08-22
> **Responses 2/2**
>
> > [Question]: L112: "manually select the attribute dimensions of the highest quality that best suit the dataset.": how do you do this? I hope this does not produce some sort of test data feature leakage? and also not introduce additional human bias in this process?
>
> We appreciate the reviewer's thoughtful comment. To address this concern, we would like to clarify that during the selection process, we *did not use any test data* to avoid data feature leakage and minimize human bias.
> Furthermore, we acknowledge the potential introduction of human bias during manual selection, which can be a common issue for human-in-the-loop learning [1]. It is worth noting that for the four datasets studied in the paper, there is high agreement among human raters for attribute selection — we conduct an addition experiment over four human raters, and the Fleiss’ Kappa over the selection of raters are 0.81 (NYT), 0.683 (Amazon), 0.798 (Reddit) and 0.625 (StackExchange) — all of the scores fall into the category of “Substantial Agreement” or “Near Perfect Agreement”, and indicates that the selection can be consistent across individuals.
>
> To further mitigate the bias, it is crucial to involve multiple team raters, cross-validation, and additional review to achieve a balanced and unbiased selection of attribute dimensions that contribute positively to the dataset's quality and diversity. In the **Appendix A of the revised manuscript**, we provide an explanation of our selection process and discuss the potential bias issue.
>
> > More instruction learning/prompting references need to be discussed, since the data generation approach is primarily based on instructing LLM to generate data.
>
> We appreciate the reviewer's feedback. In the **section 2 of the revised manuscript**, we added an additional paragraph to provide a comprehensive overview of other prompting/ instructional learning references. We hope that our revised version will help our readers better understand the foundation of our approach and the strategies we employ to achieve meaningful and coherent data generation.
>
> > For data generation, the following paper is relevant and needs to be discussed https://arxiv.org/abs/2201.05955
>
> Thanks for pointing out this relevant paper! It also considers human-AI collaboration for creating more challenging training data but requires an initial dataset and a strong task model. We have added more discussion on this paper in **section 2 “related work” of the revised manuscript**.
>
> > some model beyond ChatGPT will make it complete.
>
> We have also conducted additional experiments using the LLaMA-2 and other GPT models as generators. Please see our general response for detailed experimental results.
>
> > Since authors use ChatGPT for data generation, I hope the released license of the dataset is in compliance with OpenAI guidelines for their API usage. Authors may want to clarify this further.
>
> We appreciate the reviewer's concern regarding the dataset and ChatGPT's usage. We want to assure the reviewer that we have taken the OpenAI guidelines for API usage into careful consideration. The dataset used for training has been curated and processed in alignment with OpenAI's guidelines to ensure compliance.
>
> According to the OpenAI policy (https://openai.com/policies/terms-of-use), "Subject to your compliance with these Terms, OpenAI hereby assigns to you all its right, title and interest in and to Output." It's also worth noting that our licensing approach parallels that of various open-source projects, including Alpaca (https://github.com/tatsu-lab/stanford_alpaca#data-release). These projects, which also utilize data generated by GPT, share a similar licensing perspective as ours.
>
> We also notice that the OpenAI policy states that `You may not use the output from the Services to develop models that compete with OpenAI`, It's important to clarify that our work is not intended to rival OpenAI. Our primary focus is centered on analyzing and enhancing performance in specific downstream tasks rather than engaging in competition with OpenAI's offerings.
>
> Thanks again for your feedback! Please let us know if you have any further questions.
>
> ## Reference:
>
> [1] Casper, Stephen, et al. "Open Problems and Fundamental Limitations of Reinforcement Learning from Human Feedback." arXiv preprint arXiv:2307.15217 (2023).

---

### Official Review · Reviewer_t81z · 2023-07-28
**Review for paper "Large Language Model as Attributed Training Data Generator: A Tale of Diversity and Bias"**

**Rating:** 6
**Confidence:** 3
**Correctness:** Yes
**Clarity:** Yes

**Strengths:**

1. This paper evaluated different data prompting methods and utilize generated data for model training. The experiments demonstrated utilizing AttrPrompt to generate synthetic training data can improve model performance in general.
2. The topic raised in this paper is interesting and the paper is well written.
3. The proposed approach may have more applications and impact.

**Additional Feedback:**

NA

**Documentation:**

Yes

**Limitations:**

1. The appendix link are not working properly, it link to incorrect place.
2. It will be better to add diversity metric to quantify the AttrPrompt performance over SimPrompt and Gold.
3. Since there are multiple ways to design prompt, it will be better to explore more about the prompting ways and add more baselines in evaluation part.

**Opportunities For Improvement:**

1. The appendix link are not working properly, it link to incorrect place.
2. It will be better to add diversity metric to quantify the AttrPrompt performance over SimPrompt and Gold.
3. Since there are multiple ways to design prompt, it will be better to explore more about the prompting ways and add more baselines in evaluation part.

**Relation To Prior Work:**

Yes

**Summary And Contributions:**

This paper evaluated training data generation with diversely attributed prompts (e.g., specifying attributes like length and style), which have the potential to yield diverse and attributed generated data. This paper focuses on datasets with high cardinality and diverse domains, wherein the authors demonstrate that attributed prompts outperform simple class-conditional prompts in terms of the resulting model’s performance.
1. This paper evaluated different data prompting methods and utilize generated data for model training. The experiments demonstrated utilizing AttrPrompt to generate synthetic training data can improve model performance in general.
2. The topic raised in this paper is interesting and the paper is well written.
3. The proposed approach may have more applications and impact.

---

> ### Author Response · Authors · 2023-08-22
>
> Thanks for your thoughtful feedback! We discuss your raised points as follows:
>
>
> > The appendix link are not working properly, it link to incorrect place.
>
> Thank you for pointing out the issue. It is mainly because we manually split the main paper and the appendix. Please directly refer to the corresponding section of the Appendix for details. Sorry for the inconvenience.
>
> > It will be better to add diversity metric to quantify the AttrPrompt performance over SimPrompt and Gold.
>
> Thanks for this useful suggestion. In our original experiments, we have calculated the vocabulary coverage (in table 5) to quantify the diversity of AttrPrompt and SimPrompt. We have added two additional metrics, namely *average pairwise cosine similarity* (APS) and *Inter-sample N-gram Frequency (INGF)*  (in paper [1] as mentioned by the reviewer rTXq) to further quantify the diversity.
> For APS, the *lower* stands for better diversity. For INGF, the *higher* stands for better diversity.  The result is shown as follows:
>
>
> |            |      |    **NYT**   |      |                 |  |    **Amazon**   |      |                 |
> |------------|--------------|------|------|-----------------|-------------|------|------|-----------------|
> |            | Inter-Class APS  | Intra-Class APS | All Sample APS | N-gram Frequency | Inter-Class APS | Intra-Class APS | All Sample APS  | N-gram Frequency |
> | Simprompt  | 0.101        | 0.568| 0.135| 5277.2          | 0.207       | 0.620| 0.241| 2266.5         |
> | Attrprompt | 0.159        | 0.474| 0.182| 6688.6          | 0.225       | 0.483| 0.246| 2605.5         |
> | Gold       | 0.098        | 0.358| 0.122| 7618.1          | 0.101       | 0.251| 0.114| 4992.1       |
>
> |            |    |    **Reddit**  |      |                 |  |   **StackExchange**   |      |                 |
> |------------|--------------|------|------|-----------------|-------------------|------|------|-----------------|
> |            | Inter-Class APS | Intra-Class APS | All  | N-gram Frequency | Inter-Class APS    | Intra-Class APS | All Sample  APS | N-gram Frequency |
> | Simprompt  | 0.173        | 0.818| 0.201| 2697.8          | 0.282             | 0.804| 0.302| 2259.8          |
> | Attrprompt | 0.106        | 0.474| 0.122| 3994.5          | 0.105             | 0.375| 0.114| 2464.3          |
> | Gold       | 0.044        | 0.261| 0.054| 9079.6          | 0.056             | 0.196| 0.063| 5492.4          |
>
> Overall, we observe that under these two new metrics, AttrPrompt achieves better diversity than Simprompt, which further justifies our claims. **We have added additional experimental results and discussions in table 6, section 4.1 in the revised manuscript.**
>
>
> > Since there are multiple ways to design prompt, it will be better to explore more about the prompting ways and add more baselines in evaluation part.
>
> Following your suggestions, we have added one additional baseline named *MetaPrompt* [2] for comparison. *Please see our general response for details.* In addition, we believe that the success of attributed prompt would inspire advanced prompt techniques that leverage attributes for data generation, which we would leave to future work.
>
>
> Thanks again for your feedback! Please let us know if you have any further questions.
>
> ## References
>
> [1] Mishra et al. "Dqi: Measuring data quality in nlp." arXiv preprint arXiv:2005.00816 (2020).
>
> [2] Reynolds and McDonell. "Prompt programming for large language models: Beyond the few-shot paradigm." CHI 2021.

---

### Author Response · Authors · 2023-08-22
**General Response to All Reviewers (1/2)**

We would like to sincerely thank all reviewers for the insightful feedback. Your expert knowledge has helped us strengthen the manuscript significantly.  Here, we also provide a response to some general questions:


## Additional Experiments on LLM Generators

We added additional experimental results using other instruction-finetuned GPT models as the generator, namely `text-ada-001`, `text-babbage-001`, `text-curie-001`, and `GPT-4` (due to budget constraints, we only generate a subset with 10% size of the original dataset).
|    NYT        |     text-ada-001     | text-babbage-001 | text-curie-001   | ChatGPT | GPT-4 (10% size) |
|------------|---------|---------|---------|---------|---------|
| Simprompt  | 72.96   | 75.92   | 80.65   | 76.34   | 79.66   |
| Attrprompt | 75.85   | 78.67   | 82.63   | 82.26   | 81.69   |

|     Amazon       |       Text-ada-001      | text-babbage-001 | text-curie-001   | ChatGPT | GPT-4 (10% size) |
|------------|---------|---------|---------|---------|---------|
| Simprompt  | 45.98   | 52.26   | 55.12   | 54.56   | 57.06   |
| Attrprompt | 57.50    | 62.28   | 64.70    | 65.87   | 63.18   |

|     Reddit       |      Text-ada-001      | text-babbage-001 | text-curie-001     |ChatGPT | GPT-4 (10% size) |
|------------|---------|---------|---------|---------|---------|
| Simprompt  | 46.50   | 50.01   | 51.50    | 53.81   | 52.61   |
| Attrprompt | 51.40   | 56.58   | 61.80    | 63.10    | 58.44   |

|    StackExchange        |      Text-ada-001      | text-babbage-001 | text-curie-001     | ChatGPT | GPT-4 (10% size) |
|------------|---------------|---------|---------|---------|---------|
| Simprompt  | 38.50          | 40.55   | 42.16   | 40.83   | 40.55   |
| Attrprompt | 40.72         | 42.07   | 44.15   | 47.42   | 47.74   |

Overall, we observe that under all settings, our model outperforms the direct baseline SimPrompt with a great margin. Besides, the performance is generally better with larger models, as they often have better instruction-following capabilities. In addition, an interesting finding is that for SimPrompt (but not for AttrPrompt), the average performance of using `ChatGPT` is worse than `text-curie-001`. This finding suggests that straightforward class-dependent prompts might not exploit the capabilities of LLMs as effectively as our proposed approaches. **These results are further discussed in Figure 5, section 5.4 in the revised manuscript.**

For *non-GPT-series* model, we find that it could be time-consuming to run experiments due to our hardware constraint — generating training data using the recently released LLaMA-2 [1] can cost more than 3 days for one dataset. Despite the resource constraint, we have tried our best to show the result of using LLaMA-2-7b as the generator for the NYT dataset as:

|   Method       | LLama2   |
|----------|----------|
| Simprompt| 79.86    |
| Attrprompt| 81.53   |

We will try to complete experiments on LLaMA2-7b for other datasets.

Overall, we hope these additional results could justify the efficacy of Attrprompt over our direct baseline Simprompt.


## References

[1] Touvron, Hugo, et al. "Llama 2: Open foundation and fine-tuned chat models." arXiv preprint arXiv:2307.09288 (2023).

---

> ### Author Response · Authors · 2023-08-22
> **General Response to All Reviewers (2/2)**
>
> ## Additional Baseline on Prompting LLMs for Training Data Generation
>
> We have added on additional baseline named *MetaPrompt* [2], which uses a large language model itself to optimize the input prompt. The performance of MetaPrompt is in the following table:
>
>
> | **Method**          | **NYT** (Acc.) | **NYT** (F1) | **NYT** (Price/1k) | **Amazon** (Acc.) | **Amazon** (F1) | **Amazon** (Price/1k) | **Reddit** (Acc.) | **Reddit** (F1) | **Reddit** (Price/1k) | **StackExchange** (Acc.) | **StackExchange** (F1) | **StackExchange** (Price/1k) |
> |---------------------|----------------|--------------|--------------------|-------------------|-----------------|-----------------------|------------------|----------------|----------------------|--------------------------|------------------------|----------------------------|
> | SimPrompt            | 75.47          | 76.22        | 0.76              | 57.34             | 56.96           | 0.77                  | 53.48            | 53.81          | 0.65                  | 42.88                    | 41.30                  | 0.69                       |
> | MetaPrompt          | 79.58          | 79.83        | 0.87              | 56.35             | 55.98           | 0.84                  | 54.61            | 54.30          | 0.74                  | 44.81                    | 44.02                  | 0.83                       |
> | AttrPrompt | 81.30          | 82.26        | 1.05              | 66.08             | 65.65           | 0.87                  | 63.33            | 63.10          | 0.84                  | 48.99                    | 47.42                  | 0.90                       |
>
> We have also updated the experimental results in Table 7, 8 and 13 in the updated manuscript. We observe that although it uses additional techniques to polish the input prompts, it does not directly address the limited diversity issue, which is one unique challenge for the training data generation task. As a result, Attrprompt still outperforms MetaPrompt with an **average gain of 5.65%** over four datasets. We have added more discussion in **section 5.1 in the revised manuscript**.
>
>
> ## Scope of this paper
>
>  This year's call for papers (https://neurips.cc/Conferences/2023/CallForDatasetsBenchmarks) has extended the track's scope to include **"all work on data-centric machine learning research (DMLR), covering ML datasets and benchmarks as well as algorithms, tools, methods, and analyses for working with ML data."** Our proposed data generation approach using LLMs and associated empirical study aligns with the track's objectives, particularly in "Data-centric AI methods and tools" and "Data generators." We also release the datasets we generate as new resources. We hope this clarifies our relevance within the track.
>
>
> Thanks for all the reviewers again! We hope our responses have addressed your concerns. Please let us know if you have any further questions, and we are happy to discuss further.
>
> ## References
>
> [2] Reynolds and McDonell. "Prompt programming for large language models: Beyond the few-shot paradigm." CHI 2021.

---

### Author Response · Authors · 2023-08-22
**List of the Revisions for the Manuscript**

We would like to thank all reviewers for the valuable feedbacks. Your valuable suggestions have been thoroughly considered and incorporated into the paper. Notably, to emphasize these revisions, we use the blue color to highlight the updated content throughout the revised manuscript. While answering detailed questions of each review, we also list a summary of the updates:

- We discussed more related works on LLM for data generation as well as discrete prompt optimization in Section 2.
- We added additional metrics for diverse metrics calculation in Table 6, Section 4.1.
- We added one additional baseline MetaPrompt [1] in Section 5 with additional results exhibited in Table 7, 8 and 13. The prompt format and generated contents are listed in Appendix L.
- We provided additional performance comparison using different LLMs as data generator, shown in Figure 5, Section 5.4.
- We provided explanations on the potential limitations as well as future works on attribute selection in Appendix A.
- We provide one additional ablation study result for removing the attribute selection module in Table 14, Appendix E.2.
- We added version information for the released dataset.

We hope the revised manuscript could better resolve the concerns proposed by reviewers.

## References

[1] Reynolds and McDonell. "Prompt programming for large language models: Beyond the few-shot paradigm." CHI 2021.

---

### Decision · Program_Chairs · 2023-09-22

**Decision:**

Accept (Poster)

**Comment:**

The authors investigates the effect of diversely attributed prompts on LLMs for generating training data, in contrast to simple class-conditional prompts (simple prompts) which are widely used in practice. The authors suggest that the simple prompts could introduce bias and limit diversity. Through experiments, the paper demonstrates that attributed prompts outperform simple prompts in terms of trained model performance, bias mitigation, and data efficiency.

Reviewers hold the consensus that the new ways of writing attributed prompts seem to be 1) data-efficient and promising, 2) address challenges such as bias and diversity in LLM-generated training data in a timely manner, and 3) analyses are informative for understanding the data-generation process.

Reviewers requested additional metrics, different types of comparisons, more baselines, and suggestions for other LLMs as data generators. It seems that the authors have faithfully responded to these in their rebuttals and reflected them already in the manuscript.